# RESEARCHARCADE: GRAPH INTERFACE FOR ACADEMIC TASKS

## ABSTRACT

Academic research generates diverse data sources. As researchers increasingly use machine learning to assist research tasks, a crucial question arises: *Can we build a unified data interface to support the development of machine learning models for various academic tasks?* Models trained on such a unified interface can better support human researchers throughout the research process and eventually accelerate knowledge discovery. In this work, we introduce RESEARCHARCADE, a graph-based interface that connects multiple academic *data sources*, unifies *task definitions*, and supports a wide range of *base models* to address key academic challenges. RESEARCHARCADE utilizes a coherent multi-table format with graph structures to organize data from different sources, including academic corpora from ArXiv and peer reviews from OpenReview, while capturing information with multiple modalities, such as text, figures, and tables. RESEARCHARCADE also preserves temporal evolution at both the manuscript and community levels, supporting the study of paper revisions as well as broader research trends over time. Additionally, RESEARCHARCADE unifies diverse academic task definitions and supports various models with distinct input requirements. Our experiments across six academic tasks demonstrate that combining cross-source and multi-modal information enables a broader range of tasks, while incorporating graph structures consistently improves performance over baseline methods. This highlights the effectiveness of RESEARCHARCADE and its potential to advance research progress.

## 1 INTRODUCTION

Academic research represents a pinnacle of human knowledge discovery. Diverse research tasks such as forecasting research trends and debugging scientific papers (Sundar et al., 2024; Lin et al., 2024; Tian et al., 2025; Feng et al., 2024; 2025a; Liu et al., 2025) demand access to comprehensive data from multiple sources. To accomplish these tasks, various models are employed. These complexities raise an important research question: *Can we build a unified data interface to support the development of machine learning models for various academic tasks?*

Building such an interface for research tasks is challenging. In terms of data, firstly, academic data is sourced from diverse platforms such as ArXiv and OpenReview, encompassing complex relationships among entities like authors, papers, citations, and reviews. This requires a flexible framework capable of managing highly relational data. Secondly, the data representations themselves span multiple modalities—from textual content to visual and tabular data. Holistically integrating these varied representations is a significant challenge. Additionally, the dynamic and ever-evolving nature of academic data further complicates the task, as continuous growth and maintenance of the framework are required to keep pace with ongoing research developments. In terms of tasks, defining different academic tasks demands significant effort in data preprocessing and task formulation. In terms of models, different types of models require distinct interfaces. For example, Large Language Models (LLMs) require text-based data as input, while Graph Neural Networks (GNNs) utilize graph-structured data.

Despite existing efforts to benchmark scientific research, developing a unified and dynamic representation of research activities remains an open challenge. While existing academic datasets have systematically collected and organized academic data (Kang et al., 2018; Lo et al., 2019), they

mainly focus on single-source data, such as academic corpora or peer reviewing conversations. Although multi-modal data (e.g., figures and tables within scientific papers) have been incorporated to construct valuable datasets (Xia et al., 2024; Tian et al., 2025), these approaches do not fully exploit the multi-modal relations among different data types. Recent works have used graphs to model academic data and define academic tasks (Li & Tajbakhsh, 2023; Zhang et al., 2024). However, each academic task is still formulated individually, requiring repetitive developmental efforts.

In this paper, we propose RESEARCHARCADE, a graph-based interface that links diverse academic *data sources*, with unified *task definitions*, and supports a large variety of *base models* to solve valuable academic tasks. Overall, RESEARCHARCADE exhibits four core features that make it ideal for solving academic tasks: *Multi-Source*, *Multi-Modal*, *Highly Structural and Heterogeneous*, and *Dynamically Evolving*. RESEARCHARCADE integrates academic data from multiple sources, including research papers from ArXiv and peer reviews with revisions from OpenReview, while collecting multi-modal information, including text, figures, and tables. These distinct entities are organized in a coherent multi-table format, with selected tables designated as nodes and edges, enabling RESEARCHARCADE to efficiently handle the highly relational and heterogeneous data as graphs within academic communities. Moreover, RESEARCHARCADE models academic evolution at two scales: microscopically, it preserves paper revisions with temporal information to track individual manuscript development, and macroscopically, its extensible framework enables continuous data incorporation, supporting analysis of research trends over time. Furthermore, we unify diverse academic tasks within the academic graphs in RESEARCHARCADE, enabling straightforward formulation of new tasks across both predictive and generative paradigms. Additionally, the structured knowledge in RESEARCHARCADE can be easily exported to standardized formats, such as CSV and JSON, facilitating integration with various models, including LLMs and GNNs.

To demonstrate the key advantages of RESEARCHARCADE, we define six academic tasks: figure/table insertion, paragraph generation, revision retrieval, revision generation, acceptance prediction, and rebuttal generation. Extensive experiments show that models benefit from the multi-source, multi-modal, heterogeneous, and dynamic information in RESEARCHARCADE.

Overall, our key contributions include: First, RESEARCHARCADE enables diverse task definitions by integrating multiple data sources, multi-modal information, and supporting the inclusion of temporal and up-to-date data. Second, RESEARCHARCADE facilitates the academic task solving by unifying the task formulations and supporting the training of various models. Finally, RESEARCHARCADE shows that incorporating graph structures consistently enhances model's performance compared to baseline approaches.

## 2 RELATED WORK

**Academic data as graphs**. Existing research on academic graphs employs various decompositions on academic data. OAG-BENCH (Zhang et al., 2024) defines nodes such as authors, papers, and affiliations, modeling academic communities as heterogeneous graphs. UNARXIVE (Saier et al., 2023) and DOCGENOME (Xia et al., 2024) create finer-grained graphs by further decomposing academic corpora into paragraphs. UNARXIVE focuses on paragraph-level citations and DOCGENOME considers multi-modal elements (e.g., figures and tables). In RESEARCHARCADE, we integrate all these heterogeneous entities and extend them with comprehensive and elaborated graphs.

**Dynamic modeling of academic data**. Academic data are evolving dynamically, and their evolution is broadly classified into two parts: community research trends and individual manuscript evolution. For inter-paper evolution, Gollapalli & Li (2015) analyzes twenty years of ACL and EMNLP proceedings using topic distributions to trace venue convergence and divergence, while Tian et al. (2023) models scientific subcommunity evolution as event prediction, detecting growth, splits, and merges in collaboration graphs. For intra-paper evolution, Kuznetsov et al. (2022); D'Arcy et al. (2024) align revisions at the sentence level while Jourdan et al. (2025) focuses on the paragraph level. In RESEARCHARCADE, both evolutions are modeled simultaneously.

**Solving academic tasks with deep learning**. Various deep learning models are utilized to solve the academic tasks. Yu et al. (2025a;b) conducted end-to-end scientific discovery based on LLMs. Zhang et al. (2024) leveraged CNNs, GNNs, and LLMs to solve diverse academic tasks. However,

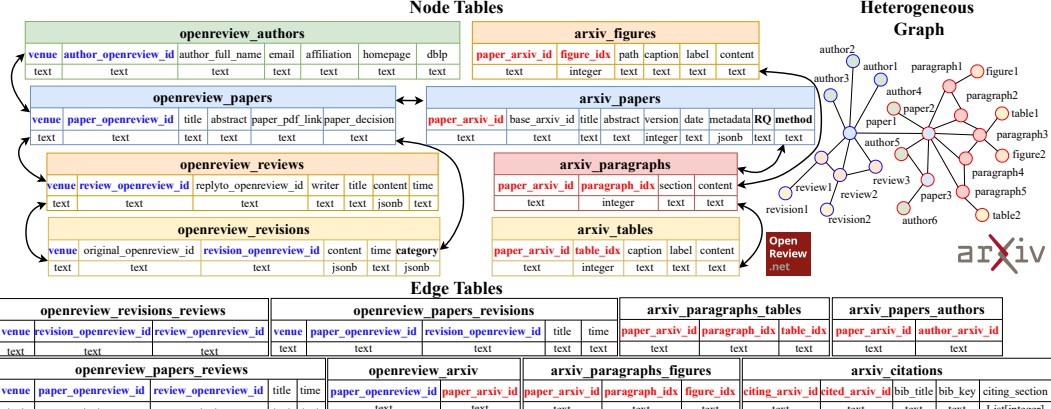

Figure 1: **RESEARCHARCADE uses a multi-table format with graph structures to collect data from different sources with multiple modalities.** Tables are classified into node tables (colored) or edge tables (black and white). The blue (denoting the OpenReview part) or red (denoting the ArXiv part) columns represent the unique identification of each node or edge, and the remaining columns represent the features of the nodes or edges. And the black bold columns are generated by LLM. The conversion from the multiple tables to heterogeneous graphs is straightforward.

their efforts are scattered and require specialized data for different models. RESEARCHARCADE offers a general graph interface to unify input data and task definitions for academic tasks.

# 3 RESEARCHARCADE DATA DESCRIPTION

RESEARCHARCADE is an inclusive mapping of real-world research knowledge, featuring four key attributes: (1) multi-source, (2) multi-modal, (3) highly relational and heterogeneous, (4) dynamically evolving. An overview is illustrated in Figure 1, with further details in Appendix Figure 3.

## 3.1 MULTI-SOURCE & MULTI-MODAL

RESEARCHARCADE is primarily sourced from computer science papers in ArXiv and available peer review data from conferences in OpenReview. Beyond text-based data, RESEARCHARCADE also integrates multi-modal data (e.g., figures and tables), supporting more complex multi-modal tasks.

**ArXiv**: RESEARCHARCADE includes 66,918 papers from ArXiv across 11 scientific fields, comprising 569,501 sections, 8,014,095 paragraphs, 876,636 figures, 324,648 tables. Relevant connections between these entities are also captured by RESEARCHARCADE. Detailed statistics are provided in Table 5, and the procedure of data collection is in Appendix A.2.1. Research Arcade also supports continuous crawling, which updates the ArXiv dataset on a routine basis (e.g. weekly, daily). The detailed description is included in Appendix A.2.2

**OpenReview**: RESEARCHARCADE also includes data from OpenReview, which comprises 57,278 submissions from ICLR, NeurIPS, ICML, and EMNLP conferences, contributed by 189,038 authors. We have also explored CVPR, ECCV, AAAI, IJCAI, ACL, and NAACL conferences, but their peer review data are unavailable. In addition, the corresponding 884,875 reviews and 54,467 submission revisions during the rebuttal process are included. These entities are enriched with valuable connections. Detailed statistics are given in Table 6 to Table 9, and the step-by-step data collection procedure is described in Appendix A.2.3.

**Connect ArXiv and OpenReview**: Connecting the data from the ArXiv and OpenReview contributes to more comprehensive academic graphs, allowing the definition of more diverse academic tasks. To achieve this goal, each submission in OpenReview is associated with its corresponding paper in ArXiv based on the title. Note that 25,969 (about 45.34%) submissions from OpenReview are successfully connected to papers from ArXiv. The statistics are shown in Table 10 to Table 13.

**LLM-Generated Content**: To facilitate the use of RESEARCHARCADE for more academic tasks (e.g., Contradiction Detection, Theory Synthesis), we used Llama-3.1-70B-Instruct (Grattafiori

Figure 2: **RESEARCHARCADE unifies the academic task definitions in a two-step scheme**: (i) Label: Identify the task's target entity and assign its attribute as label; (ii) Input: Retrieve the target entity's neighborhood to construct an academic graph that supports task solving.

et al., 2024) for data preprocessing. The submission revisions are classified according to the categories outlined in Jourdan et al. (2025). The detailed descriptions of these categories are provided in Table 14. Additionally, we apply the same model to generate research question summaries and method descriptions for ArXiv papers. The specific prompts used are listed in Table 15 to Table 17.

## 3.2 HIGHLY RELATIONAL AND HETEROGENEOUS

Research activities in academic communities are modeled by interactions among typed entities. RESEARCHARCADE stores data in a multi-table node–edge schema, consisting of node tables and edge tables, which directly map to heterogeneous graphs. An illustration is shown in Figure 1.

Using data from ArXiv, RESEARCHARCADE constructs a two-scale graph representation of the literature. At the intra-paper level, each paper is decomposed into a paragraph-scale content graph including paper, paragraphs, figures, and tables nodes, linked by typed edges (e.g., paper-paragraph, paragraph-figure/table). At the macro inter-paper level, we include authors, subject categories, and citation links, adding edges for authorship, category assignment, and paper-to-paper citations.

The academic graphs built on data from OpenReview mainly model the academic activities that happen during the peer review process. It encompasses diverse types of nodes, such as papers, authors, paragraphs, reviews, and revisions. Some key relationships are also included: the authorship, which connects papers and authors; the comment-under-paper relation, which connects papers and reviews; the revision-of-paper relation, which connects papers and revisions; the revision-caused-by-review relation, which connects reviews and revisions, etc.

## 3.3 DYNAMICALLY EVOLVING

As the academic community continuously evolves, RESEARCHARCADE records temporal information (e.g., paper upload dates and paper revision timestamps), enabling a realistic simulation of scholarly dynamics. This includes tracing the evolution of research trends and modeling paper updates driven by the rebuttal process. Moreover, RESEARCHARCADE can be continuously updated to reflect the ongoing development in the academic community.

## 4 ACADEMIC TASKS ON RESEARCHARCADE

Defining different academic tasks often requires repetitive work, such as data collection, cleaning, and task specification. With RESEARCHARCADE, these tasks can be unified and conveniently defined on our academic graphs.

## 4.1 ACADEMIC GRAPH AS A HETEROGENEOUS GRAPH

A heterogeneous graph can be defined as $\mathcal{G} = (\mathcal{V}, \mathcal{E})$, where each node $v \in \mathcal{V}$ and each edge $e \in \mathcal{E}$ is assigned a type through mapping functions. Specifically, the node type is defined by $\tau(v) : \mathcal{V} \to \mathcal{C}$,

Table 1: **Summary of six academic tasks studied with RESEARCHARCADE.** Abbreviations include "or": openreview, "ar": ArXiv, CE: Cross-entropy Loss, BCE: Binary Cross-entropy Loss.

| Task | Target Entity (Step 1) | Neighborhood (Step 2) | Loss | Type |
|------|------------------------|------------------------|------|------|
| Citation Prediction | ar_citation edge: content of the citing paragraph | ar_section nodes, ar_paragraph nodes, ar_figure nodes, ar_table nodes, ar_paragraph nodes, ar_paper nodes | CE | Predictive |
| Paragraph Generation | ar_paragraph node: Textual content of the paragraph | ar_paragraph nodes, ar_table nodes, ar_figure nodes, ar_citation edges | SFT | Generative |
| Revision Retrieval | or_revision node: Index list of modified paragraphs | or_paragraph nodes from the original paper, or_review nodes | InfoNCE | Predictive |
| Revision Generation | or_paragraph node: Textual content of the revised paragraph | or_paragraph node of the original paper, or_review nodes | SFT | Generative |
| Acceptance Prediction | or_paper node: Paper decision | or_paper nodes, ar_paper nodes, ar_paragraph nodes, ar_figure nodes, ar_table nodes | BCE | Predictive |
| Rebuttal Generation | or_review node: Textual content of the author's response | or_review node of the official review being replied to, ar_paper node, ar_paragraph nodes, ar_figure nodes, ar_table nodes | SFT | Generative |

and the edge type is defined by $\phi(e) : \mathcal{E} \to \mathcal{D}$, where $c \in \mathcal{C}$ and $d \in \mathcal{D}$ represent the set of node types and the set of edge types. An edge $e$ connecting a pair of nodes is denoted as $e = (v, u)$.

Data from RESEARCHARCADE can be represented as an academic graph $\mathcal{G} = (\mathcal{V}, \mathcal{E})$, which is heterogeneous. In this context, each node $v \in \mathcal{V}$ corresponds to a row in the node table, while each edge $e$ corresponds to a row in the edge table. Furthermore, each node table $V_c$ is associated with a unique node type $c$, and each edge table $E_d$ is linked to a unique edge type $d$.

## 4.2 UNIFIED ACADEMIC TASK DEFINITION

As is shown in Figure 2, RESEARCHARCADE unifies the academic task definitions in the following two steps: (1) identifying the target entity and (2) retrieving the neighborhood of the target entity.

**Step 1: Identifying the target entity of an academic task.** The target entity is either a node $v$ or an edge $e$, with attributes that define the labels for the task. Let $t$ denote the target entity with attributes $\mathbf{a}_t$. Its certain attributes, denoted as $\mathbf{y}_t \subseteq \mathbf{a}_t$, are the labels implied in the task.

**Step 2: Retrieving the neighborhood of the target entity.** To support the academic task solving, the multi-hop neighborhood of the target entity $t$ is retrieved, constructing an academic graph $\mathcal{G}_t$ centered at $t$. The one-hop neighborhood $\mathcal{N}_t^{(1)}$ of $t$ consists of entities directly connected to $t$. If $t \in \mathcal{V}$, then $\mathcal{N}_t^{(1)} = \{k \,|\, k \in \mathcal{V}, (t, k) \in \mathcal{E}\}$. If $t \in \mathcal{E}$, then $\mathcal{N}_t^{(1)} = \{k, u \,|\, k, u \in \mathcal{V}, t = (k, u)\}$. For $i > 1$, the $i$-hop neighborhood is defined as $\mathcal{N}_t^{(i)} = \{k \,|\, k \in \mathcal{V}, k' \in \mathcal{N}_t^{(i-1)}, (k, k') \in \mathcal{E}\}$, which extends the $(i-1)$-hop neighborhood by one additional hop. Hence, the academic graph is constructed as $\mathcal{G}_t = (\mathcal{V}_t, \mathcal{E}_t)$, where $\mathcal{V}_t$ contains nodes in the multi-hop neighborhood of $t$, and $\mathcal{E}_t$ represents the edges between these nodes. Thus, an academic task is defined as follows:

$$f_\theta(\mathcal{G}_t) \to \mathbf{y}_t, \tag{1}$$

where $f_\theta$ represents a model with parameters $\theta$. Furthermore, the academic tasks are broadly classified into predictive and generative tasks. If the label $\mathbf{y}_t$ is from a limited set of possible outcomes, this task is categorized as a predictive task; If the label $\mathbf{y}_t$ is in an open-ended output space, this task is categorized as a generative task. **For predictive tasks**, models (specified in Section 5.1) are considered as MLP-based, Embedding-based, GNN-based, or GWM-based, where the GWM framework efficiently integrates graph-structured data with LLM (Feng et al., 2025b). The training loss varies across different predictive tasks. **For generative tasks**, models are primarily based on LLMs. Supervised fine-tuning (SFT) is used for training, with the loss defined as follows:

$$\mathcal{L}_{\text{SFT}}(\theta) = -\frac{1}{\sum_{t=1}^{T} L_t} \sum_{t=1}^{T} \sum_{i=1}^{L_t} \log p_\theta\big(y_{t,i} \,|\, y_{t,<i}, \mathcal{G}_t\big), \tag{2}$$

where $L_t$ is the length of $\mathbf{y}_t = [y_{t,1}, ..., y_{t,L_t}]$, and $\log p$ is the log-likelihood. In this paper, six academic tasks are defined to demonstrate the four key features of RESEARCHARCADE. Table 1 summarizes the tasks under the two-step scheme with detailed task definitions in Appendix A.3.

#### 4.2.1 CITATION PREDICTION

Citation prediction requires the model to identify the appropriate paper to cite for a given paragraph, reflecting real-world needs like reference recommendation. While previous work (Arthur Brack, 2021) focuses on paper-level citation, we conduct the task at the paragraph level using the fine-grained academic graphs in RESEARCHARCADE. We formulate this task as a multi-class classification problem. Given an academic graph $\mathcal{G}_t$ that contains all paragraphs and existing citations of a paper, along with a candidate paragraph, the model predicts the paper $\hat{y}_t$ that should be cited by the target paragraph $\mathbf{y}_t$. The ground truth $\mathbf{y}_t$ corresponds to the paper that was actually cited in the target paragraph. Here, we optimize the model using the contrastive cross-entropy loss:

$$\mathcal{L}_{\text{CE}}(\theta) = -\log \frac{\exp\left(\text{sim}(h_t, z_{\mathbf{y}_t})/\tau\right)}{\sum_{j=1}^{M} \exp\left(\text{sim}(h_t, z_j)/\tau\right)}, \tag{3}$$

where $\theta$ denotes the model parameters, $h_t$ is the embedding of the target paragraph, $z_j$ is the embedding of the $j$-th candidate cited paper, $M$ is the total number of candidate cited papers, $\mathbf{y}_t$ is the index of the ground-truth cited paragraph, $\tau$ is a temperature hyperparameter, and $\text{sim}(\cdot, \cdot)$ denotes cosine similarity between $\ell_2$-normalized embeddings. This objective encourages the model to assign higher similarity to the true cited paper than to other candidate papers.

#### 4.2.2 ACADEMIC TASK 2: PARAGRAPH GENERATION

Understanding how to generate specific paragraphs within their proper context in academic corpora is essential for assisting scientific writing. The inherent graph structures within RESEARCHARCADE offer relational signals among paragraphs, which are valuable for models to comprehend structural dependencies within corpora. This generative task is defined as follows: given the input, an academic graph $\mathcal{G}_t$ including surrounding paragraphs, referenced figures and tables, and cited literature, generate the missing paragraph content $\hat{\mathbf{y}}_t$. The original paragraph content serves as the ground truth label $\mathbf{y}_t$. To train the LLM, SFT loss (Eq. 2) is utilized. The prompt designed to help the LLM better understand the document completion task is shown in Table 22.

#### 4.2.3 ACADEMIC TASK 3: REVISION RETRIEVAL

Identifying the precise location of revisions from reviewers' comments is essential for paper refinement. This captures intra-paper dynamics during peer review and demonstrates RESEARCHARCADE's ability to model evolving content based on graph structures. We formulate this as a top-$k$ ranking task: given an academic graph $\mathcal{G}_t$ containing paper paragraphs and reviews, predict the top-$k$ modified paragraphs $\hat{\mathbf{y}}_t$, with ground truth $\mathbf{y}_t$ denoting the actual revised paragraphs. Training employs the InfoNCE loss (He et al., 2020), which minimizes embedding distance between reviews and revised paragraphs while maximizing distance from unchanged ones:

$$\mathcal{L}_{\text{InfoNCE}}(\theta) = -\frac{1}{R} \sum_{r=1}^{R} \log \frac{\sum_{i=1}^{M^+} \exp\left(\text{sim}(q_r, k_i^+)/\tau\right)}{\sum_{i=1}^{M^+} \exp\left(\text{sim}(q_r, k_i^+)/\tau\right) + \sum_{j=1}^{M^-} \exp\left(\text{sim}(q_r, k_j^-)/\tau\right)}, \tag{4}$$

where $\theta$ denotes the model parameters; $q_r$ is the model-generated embedding of the $r$-th review ($r = 1, ..., R$); $k_i^+$ and $k_j^-$ are the embeddings of the $i$-th modified and $j$-th unchanged paragraph, respectively; $M^+$ and $M^-$ are their counts; $\text{sim}(\cdot, \cdot)$ is the similarity function; and $\tau$ is the temperature in the InfoNCE loss.

#### 4.2.4 ACADEMIC TASK 4: REVISION GENERATION

Building on Section 4.2.3, this task focuses on generating quality-enhancing revisions of localized paragraphs conditioned on reviewer feedback, further demonstrating RESEARCHARCADE's dynamic evolution capability based on graph structures. Unlike previous works on revision generation (D'Arcy et al., 2024; Jourdan et al., 2025), based on the academic graphs in RESEARCHARCADE, we can conveniently retrieve the corresponding comments from the reviewer to facilitate the task. Formally, given an academic graph $\mathcal{G}_t$ containing the original paragraph and its reviews, the goal is to generate a revised paragraph $\hat{\mathbf{y}}_t$, with the actual revision $\mathbf{y}_t$ as the label. Training uses SFT loss (Eq. 2), supported by a task-specific prompt (Table 25) to guide the LLM in leveraging graph structures. Since LLMs have limited context length, reviews are first summarized using Qwen3-8B with the prompt in Table 24.

Table 2: **Promising new tasks enabled by RESEARCHARCADE for future works.**

| Task | Target Entity (Step 1) | Neighborhood (Step 2) | Loss | Type |
|------|------------------------|------------------------|------|------|
| Idea Generation | ar_paper node: Abstract | ar_citation edges, ar_paper nodes | SFT | Generative |
| Experiment Planning | ar_table node: Table text in experiment section | ar_paper node, ar_section nodes, ar_paragraph nodes, ar_figure nodes, ar_table nodes | SFT | Generative |
| Abstract Writing | ar_paper node: Abstract | ar_paper node, ar_section nodes, ar_paragraph nodes, ar_figure nodes, ar_table nodes | SFT | Generative |
| Review Generation | or_review node: Textual content of the official review | or_paper node, or_paragraph nodes | SFT | Generative |

### 4.2.5 ACADEMIC TASK 5: ACCEPTANCE PREDICTION

Predicting the acceptance of academic papers is a meaningful but challenging task. Different from previous work (Feng et al., 2025a), which focuses only on text-based academic graphs, we fuse ArXiv's comprehensive multi-modal paper graph with OpenReview's ground-truth acceptance labels and temporal information to define the task. This reflects RESEARCHARCADE's multi-source, multi-modal, and dynamically evolving nature. We design the task as a binary classification problem: given the input, an academic graph $\mathcal{G}_t$ containing papers from conferences in previous years and their corresponding paragraphs with figures and tables, predict the paper acceptance $\hat{y}_t$ (Accept or Reject) for the future year. The real paper acceptance is the label $y_t$. Binary cross-entropy loss is utilized as the training loss:

$$\mathcal{L}_{\text{BCE}}(\theta) = -\frac{1}{T} \sum_{t=1}^{T} \Big[ \mathbf{y}_t \log \hat{\mathbf{y}}_t + (1 - \mathbf{y}_t) \log(1 - \hat{\mathbf{y}}_t) \Big]. \tag{5}$$

where $\theta$ represents the model's parameters and $T$ the total number of papers.

### 4.2.6 ACADEMIC TASK 6: REBUTTAL GENERATION

Generating rebuttal responses to official reviews is critical, as response quality strongly influences paper acceptance. Based on the academic graph in RESEARCHARCADE, this task conveniently leverages textual and multi-modal information from ArXiv along with official reviews from OpenReview. Formally, given an academic graph $\mathcal{G}_t$ containing the review and its related paragraphs with figures and tables from ArXiv, the goal is to generate the author's response $\hat{\mathbf{y}}_t$, with the true response $\mathbf{y}_t$ as the label. Training uses SFT loss (Eq. 2), guided by a task-specific prompt (Table 27) to help the LLM capture graph structure and task requirements. To address token length limits, only the top-3 related paragraphs, selected via cosine similarity between review and paragraph embeddings using Qwen3-Embedding-0.6B, are included.

### 4.3 PROMISING NEW TASKS ENABLED BY RESEARCHARCADE

The versatility of RESEARCHARCADE extends beyond the tasks defined above, supporting additional stages of the research pipeline such as idea brainstorming, experiment planning, scientific writing, and peer reviewing—core activities in the academic process. These promising new tasks are illustrated in Table 2, with detailed specifications provided in Appendix A.4.

## 5 EXPERIMENT

### 5.1 EXPERIMENT SETUP

**Dataset**: We conduct experiments based on a subset of data in RESEARCHARCADE. For data from ArXiv, we mainly focus on papers in the Computer Science field and published within the last two years. For data collected from OpenReview, we primarily focus on the ICLR conferences within the past five years. Further detailed information is provided in Appendix A.5.

**Base Models**: To demonstrate the compatibility of RESEARCHARCADE with diverse models, experiments are conducted across various base models.

**(1) Embedding model (EMB)**: Considering the relatively long token input for our academic tasks, we utilize Longformer (Beltagy et al., 2020), a model designed for processing long documents.

**(2) Graph neural network (GNN)**: Since the academic graphs constructed from our database are highly relational and heterogeneous, we consider HANConv (Wang et al., 2019), a heterogeneous graph attention neural network, as our GNN-based model.

**(3) Large language model (LLM)**: We mainly leverage Qwen3-0.6B and Qwen3-8B (Yang et al., 2025) as our LLM-based models, as they outperform models with an approximate number of parameters and are comparable to larger models in various evaluation tasks. We also validate our tasks on GPTOSS-120B (Agarwal et al., 2025), a larger state-of-the-art model.

**(4) Graph world model (GWM)**: To efficiently integrate graph-structured data with LLMs, we employ the embedding-based GWM (Feng et al., 2025b). It adopts a multi-hop aggregation to perform an embedding-level message passing, yielding an enhanced graph representation, which facilitates better LLM comprehension of the graph-structured data. Qwen3-0.6B (Yang et al., 2025) is utilized as the LLM module for the GWM-based models.

**Encoders**: For the **text modality**, we represent text data as vector embeddings for integration with GNN-based and GWM-based models. Specifically, Longformer (Beltagy et al., 2020) is used for downstream GNNs, while Qwen3-Embedding-0.6B (Zhang et al., 2025) is adopted in GWM-based models to align with the Qwen3 LLM module. For the **visual modality**, LLaVA-1.5-7B (Liu et al., 2024) converts figures into textual descriptions, which are then encoded using the same text encoders. While we experimented with CLIP, our current approach is more effective and simpler to implement. This encoding framework remains flexible and can accommodate alternative multi-modal encoders.

**Evaluation Metrics**: To systematically evaluate the performance of different models on our academic tasks, different evaluation metrics are considered for each task.

**(1) Predictive Tasks**: For the top-$k$ ranking task, we report the top-5 precision, top-5 recall, and top-5 F-1 score to assess the model's performance. For the classification task, accuracy, AUC-ROC score, and Matthews correlation coefficient (MCC) are computed for evaluation.

**(2) Generative Tasks**: The semantic similarity between generated and reference answers is measured using the SBERT similarity score (Reimers & Gurevych, 2019). Lexical overlap is assessed with Rouge-L (Lin, 2004). Moreover, we leverage GPT-4o-mini (Hurst et al., 2024) to judge the clarity and the appropriateness of the output. Instead of hand-crafted evaluation metrics, we ask LLM to express pairwise preferences between the generated output and the ground truth and define the quantitative score as the preference proportion in which the generated output is preferred (including ties) over the reference.

$$\text{LLM-as-a-judge Score} = \frac{N_{\text{generated}} + N_{\text{tied}}}{N_{\text{generated}} + N_{\text{tied}} + N_{\text{truth}}}. \tag{6}$$

The specific prompt usages are shown in Appendix A.6.1.

## 5.2 EXPERIMENT RESULTS

The conclusive analysis of the experiment results is as follows, with detailed analysis of each task provided in Appendix A.7 and case studies provided in Appendix A.8.

### 5.2.1 RESEARCHARCADE IS GENERAL

Table 3 shows that RESEARCHARCADE enables diverse tasks by integrating academic corpora with multi-modal information from ArXiv and peer reviews with revisions from OpenReview, while supporting various models by converting the data into CSV or JSON formats. EMB-based, GNN-based, and GWM-based models are capable of performing predictive tasks, while LLM-based models handle generative tasks. Furthermore, the data quality in RESEARCHARCADE is validated, with trained smaller LLMs approaching the performance of larger ones. In *Revision Generation*, Qwen3-0.6B's

Table 3: **Evaluation results across six academic tasks.** Each base model follows (Backbone, Training, Hop), where Backbone is the specific model, Training is Fixed or Trained, and #-hop is the number of hops of neighbors that a model can observe. (0-hop indicates no neighbors are observed)

| Citation Prediction | | | | Paragraph Generation | | | |
|---|---|---|---|---|---|---|---|
| Model\Metric | Accuracy | AUC - ROC | MCC | Model\Metric | SBERT | Rouge-L | GPT-4o-mini |
| EMB (Longformer, Fixed, 1-hop) | 0.970 | 0.427 | 0.050 | GWM (Qwen3-0.6B, Trained, 0-hop) | 0.581 | 0.163 | 0.009 |
| GNN (HANConv, Trained, 1-hop) | **0.989** | **0.995** | 0.396 | GWM (Qwen3-0.6B, Trained, 1-hop) | 0.624 | **0.167** | 0.244 |
| GNN (HANConv, Trained, 3-hop) | 0.987 | 0.993 | **0.705** | GWM (Qwen3-0.6B, Trained, 3-hop) | 0.638 | 0.166 | 0.404 |
| GNN (HANConv, Trained, 5-hop) | **0.989** | 0.993 | **0.705** | GWM (Qwen3-0.6B, Trained, 5-hop) | **0.642** | 0.165 | 0.344 |
| Revision Retrieval | | | | Acceptance Prediction | | | |
| Model\Metric | Precision@5 | Recall@5 | F-1@5 | Model\Metric | Accuracy | AUC - ROC | MCC |
| EMB (Longformer, Fixed, 1-hop) | 0.183 | 0.154 | 0.145 | MLP (Linear, Trained, 1-hop) | 0.513 | 0.479 | 0.025 |
| GNN (HANConv, Trained, 1-hop) | **0.307** | 0.325 | **0.265** | GNN (HANConv, Trained, 1-hop) | 0.507 | 0.465 | 0.000 |
| GNN (HANConv, Trained, 3-hop) | **0.307** | 0.324 | **0.265** | GNN (HANConv, Trained, 3-hop) | **0.550** | **0.526** | **0.115** |
| GWM (Qwen3-0.6B, Trained, 1-hop) | 0.304 | 0.325 | 0.264 | GWM (Qwen3-0.6B, Trained, 1-hop) | 0.470 | 0.478 | -0.063 |
| GWM (Qwen3-0.6B, Trained, 3-hop) | 0.306 | **0.326** | **0.265** | GWM (Qwen3-0.6B, Trained, 3-hop) | 0.527 | 0.524 | 0.052 |
| Revision Generation | | | | Rebuttal Generation | | | |
| Model\Metric | SBERT | Rouge-L | GPT-4o-mini | Model\Metric | SBERT | Rouge-L | GPT-4o-mini |
| LLM (Qwen3-0.6B, Fixed, 1-hop) | 0.321 | 0.210 | 0.447 | LLM (Qwen3-0.6B, Fixed, 1-hop) | 0.604 | 0.125 | 0.011 |
| LLM (Qwen3-0.6B, Trained, 1-hop) | **0.733** | **0.554** | 0.572 | LLM (Qwen3-0.6B, Trained, 1-hop) | 0.638 | 0.131 | 0.022 |
| LLM (Qwen3-8B, Fixed, 1-hop) | 0.704 | 0.446 | 0.889 | LLM (Qwen3-8B, Fixed, 1-hop) | 0.700 | **0.154** | 0.208 |
| LLM (GPTOSS-120B, Fixed, 1-hop) | 0.669 | 0.265 | **0.999** | LLM (GPTOSS-120B, Fixed, 1-hop) | **0.703** | 0.152 | **0.884** |

SBERT similarity score and LLM-as-a-judge score (Eq. 6) improve from 0.321 to 0.733 and 0.447 to 0.572, approaching the scores of Qwen3-8B and GPTOSS-120B. And in *Rebuttal Generation*, Qwen3-0.6B's SBERT similarity score and LLM-as-a-judge score improve from 0.604 to 0.638 and 0.011 to 0.022, approaching the scores of Qwen3-8B and GPTOSS-120B.

### 5.2.2 RESEARCHARCADE MODELS DYNAMIC EVOLUTION

As shown in Table 3, RESEARCHARCADE effectively captures dynamic evolution at both the intra-paper and inter-paper levels by incorporating temporal data from ArXiv and OpenReview. The tasks of *Revision Retrieval* and *Revision Generation* highlight RESEARCHARCADE's ability to model intra-paper evolution, predicting and generating revisions that reflect the continuous development of manuscripts. In particular, the top-5 F1 scores achieved by GNN-based and GWM-based models (0.265 each) outperform the EMB-based model (0.145), underscoring the framework's effectiveness. In contrast, the *Acceptance Prediction* task reflects inter-paper evolution, aiming to identify promising papers for acceptance by learning from historical data. Here, performance was much poorer, with the best accuracy reaching only 0.55, barely above random chance. This emphasizes the inherent difficulty of predicting research trends.

### 5.2.3 RELATIONAL GRAPH STRUCTURE DELIVERS CONSISTENT GAINS

To assess the effectiveness of RESEARCHARCADE's graph-centric design, we compare graph-based models (GNN-based and GWM-based) with non-graph models (EMB-based and MLP-based) across two tasks, observing performance gains of 67%, and 7.2% in *Revision Retrieval*, and *Acceptance Prediction*, respectively, in Table 3. Multi-hop aggregation further improves performance, particularly in *Acceptance Prediction*: while 1-hop aggregation yields weak results (accuracies of 0.507 and 0.47), expanding to 3 hops raises both GNN-based and GWM-based models to 0.55, surpassing the MLP baseline (0.513). This indicates that acceptance decisions depend on higher-order context, such as venue affiliation and temporal trends, captured by multi-hop neighborhoods. The *Citation Prediction* task also investigates the impact of varying hops of aggregation. For *Citation Prediction*, although 1-hop performance is already high, expanding the neighborhood substantially improves robustness, as MCC score increases by 30.9%. However, for other tasks (e.g., *Revision Retrieval*, *Paragraph Generation*), additional hops provide little benefit or even make performance fluctuate. In *Paragraph Generation*, GPT-4o-mini score declines from 40.4% (3-hop) to 34.4% (5-hop), as larger neighborhoods may introduce irrelevant or noisy information.

### 5.2.4 MULTI-MODAL INFORMATION IS CRITICAL

Table 4 shows that incorporating figures and tables consistently enhances model performance compared to text-only baselines for the *Rebuttal Generation* and *Citation Prediction* tasks. Specifically,

Table 4: **Ablation Study on multi-model and review information.** Each base model follows (Backbone, Training, Modality), where Backbone is the specific model, Training is Fixed or Trained, Modality is with Figure & Table, with Figure, with Table, without Figure & Table, with Review, or without Review.

| Rebuttal Generation | | | | Citation Prediction | | | |
|---|---|---|---|---|---|---|---|
| Model\Metric | SBERT | Rouge-L | GPT-4o-mini | Model\Metric | Accuracy | AUC-ROC | MCC |
| LLM (Qwen3-8B, Fixed, w/o F&T) | 0.671 | 0.140 | 0.134 | GNN (HANConv, Trained, 5-hop, w/o F&T) | 0.977 | 0.990 | 0.542 |
| LLM (Qwen3-8B, Fixed, w F) | 0.692 | 0.150 | 0.178 | GNN (HANConv, Trained, 5-hop, w F) | 0.977 | 0.990 | 0.542 |
| LLM (Qwen3-8B, Fixed, w T) | 0.693 | 0.152 | 0.191 | GNN (HANConv, Trained, 5-hop, w T) | 0.980 | 0.990 | 0.564 |
| LLM (Qwen3-8B, Fixed, w F&T) | 0.700 | 0.154 | 0.208 | GNN (HANConv, Trained, 5-hop, w F&T) | 0.989 | 0.993 | 0.705 |
| Revision Retrieval | | | | Revision Generation | | | |
| Model\Metric | Precision@5 | Recall@5 | F-1@5 | Model\Metric | SBERT | Rouge-L | GPT-4o-mini |
| EMB (Longformer, Fixed, w/o R) | 0.067 | 0.043 | 0.046 | LLM (Qwen3-0.6B, Fixed, w/o R) | 0.570 | 0.401 | 0.596 |
| EMB (Longformer, Fixed, w R) | 0.183 | 0.154 | 0.145 | LLM (Qwen3-0.6B, Fixed, w R) | 0.321 | 0.210 | 0.447 |
| GNN (HANConv, Trained, w/o R) | 0.290 | 0.329 | 0.260 | LLM (Qwen3-8B, Fixed, w/o R) | 0.712 | 0.473 | 0.873 |
| GNN (HANConv, Trained, w R) | 0.307 | 0.324 | 0.265 | LLM (Qwen3-8B, Fixed, w R) | 0.704 | 0.446 | 0.889 |
| GWM (Qwen3-0.6B, Trained, w/o R) | 0.301 | 0.320 | 0.260 | LLM (GPTOSS-120B, Fixed, w/o R) | 0.672 | 0.369 | 0.924 |
| GWM (Qwen3-0.6B, Trained, w R) | 0.306 | 0.326 | 0.265 | LLM (GPTOSS-120B, Fixed, w R) | 0.669 | 0.265 | 0.999 |

both figures and tables are critical, as adding either alone yields consistent gains, while using both together gives the best performance. This suggests that the inclusion of visual and tabular data augments the model's understanding of textual content, leading to clear performance gains. For *Rebuttal Generation*, SBERT similarity score and LLM-as-a-judge score (Eq. 6) increase from $0.671$ to $0.700$ and from $0.134$ to $0.208$. Similarly, in *Citation Prediction*, the MCC increased from $0.542$ to $0.705$ when full modalities are included. These results validate RESEARCHARCADE's multi-modal design and highlight the effectiveness of its approach to encoding multi-modal information.

### 5.2.5 REVIEW INFORMATION COULD BE AMBIGUOUS

We conduct ablation studies on the review information in *Revision Retrieval* and *Revision Generation*. For *Revision Retrieval*, we remove the review information by replacing all the review content with the same prompt listed in Table 28. According to results in Table 4, incorporating specific review content delivers gains for all models. In particular, the EMB-based model exhibits a larger performance gain compared to the GNN-based and GWM-based models. The GNN-based model can exploit the review graph structure for better predictions, and the GWM-based model can further leverage the reasoning ability of its LLM module to achieve higher absolute performance. For the ablation study for *Revision Generation*, we directly prompt the model to produce a revised paragraph from the original, without incorporating the review information. Surprisingly, Qwen3-0.6B performs even worse when reviews are included, likely because the small model struggles with the longer context. And the larger models, such as Qwen3-8B and GPTOSS-120B, only show modest improvements. One reason is that many reviews lack explicit revision instructions, so the models tend to make superficial edits rather than substantial changes that would markedly improve the paragraph. In addition, some requested revisions require the author to add domain-specific content, which is difficult for the models to generate.

## 6 CONCLUSION

We introduced RESEARCHARCADE, a graph-based interface that unifies multi-source (ArXiv, OpenReview), multi-modal (text, figures, tables), and temporally evolving academic data into a coherent multi-table format. Furthermore, RESEARCHARCADE demonstrates strong scalability and supports the continuous crawling of new data on a routine basis. Building on a simple two-step scheme, (i) identify the target entity (label) and (ii) retrieve a task-specific academic graph (neighborhood), RESEARCHARCADE standardizes the definition of both predictive and generative academic tasks. RESEARCHARCADE is compatible with various models, serving as a valuable platform for studying research progress and developing models that facilitate automated scientific research. Experiments across six representative tasks show that the graph structure delivers consistent gains.

## ETHICS STATEMENT

We developed this work in accordance with the ICLR Code of Ethics and have carefully considered its broader impacts on the academic research community. Our system aims to contribute positively to research automation by providing tools for paper discovery, review assistance, and research trend analysis that could democratize access to academic insights and support researchers across different resource levels.

Potential Risks and Mitigation: We acknowledge several areas of concern regarding our academic task automation capabilities. Automated features such as paper completion and response drafting could potentially be misused for academic misconduct. We emphasize that our system is intended as a research assistance tool to augment human judgment, not replace academic thinking or writing. Additionally, our reliance on existing academic data sources (ArXiv, OpenReview) may perpetuate existing biases in publication patterns and review processes. The acceptance prediction capabilities could inadvertently influence submission strategies in ways that prioritize predicted acceptance over scientific merit rather than encouraging methodological rigor and novelty.

Data and Privacy: Our system uses exclusively publicly available academic data from ArXiv and OpenReview platforms. We respect the existing terms of use for these platforms and do not attempt to de-anonymize review processes or access private information. No human subjects are directly involved in our research process, and no additional ethical approvals were required.

Transparency and Responsible Use: We acknowledge that our graph construction and task formulation choices embed assumptions about academic workflows that may not generalize across all research domains. We encourage users to employ our system as an exploratory and assistance tool rather than for automated decision making, particularly for high-stakes academic decisions. Any research assistance provided should be subject to appropriate human oversight and verification to maintain research integrity.

## REPRODUCIBILITY STATEMENT

To ensure reproducibility of our results, we have made extensive efforts to document our methodology and provide necessary resources. Complete implementation details for our graph construction process, including multi-source data integration from ArXiv and OpenReview, are provided in A.2.1 and A.2.3. The two-step task formulation scheme is fully specified in Section 4 with concrete examples. All experimental configurations, hyperparameters, and model architectures used across the six representative tasks are detailed in 5.1 and A.5. We provide comprehensive ablation studies and statistical significance testing procedures in 5.2. Code for data processing, graph construction, model implementation, and evaluation will be made available upon publication. The constructed heterogeneous graph dataset, along with task-specific splits and evaluation protocols, will also be released to facilitate future research.

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

## A  APPENDIX

### A.1  DATA DESCRIPTION IN RESEARCHARCADE

The detailed description of RESEARCHARCADE is shown in Figure 3.

The statistical overview of data collected from ArXiv is illustrated in Table 5.

The statistical overview of data collected from OpenReview is illustrated separately in Table 6 (ICLR), Table 7 (NeurIPS), Table 8 (ICML), and Table 9 (EMNLP).

The statistical overview of openreview_arxiv table is shown in Table 10 (ICLR), Table 11 (NeurIPS), Table 12 (ICML), and Table 13 (EMNLP).

### A.2  DATA COLLECTION PROCEDURE

#### A.2.1  ARXIV

We develop a multi-stage pipeline to collect and structure papers from ArXiv. The process begins by selecting target papers using ArXiv IDs or publication date ranges. Using the ArXiv API, we download the LaTeX source files together with basic metadata. The sources are then processed through a seven-stage pipeline that converts raw LaTeX into structured graph representations.

**Stage 1: Paper Source File Downloading**  For each paper, we download and unpack the LaTeX source archive into a working directory. We also collect metadata including author names, paper categories, submission dates, paper version, and abstracts.

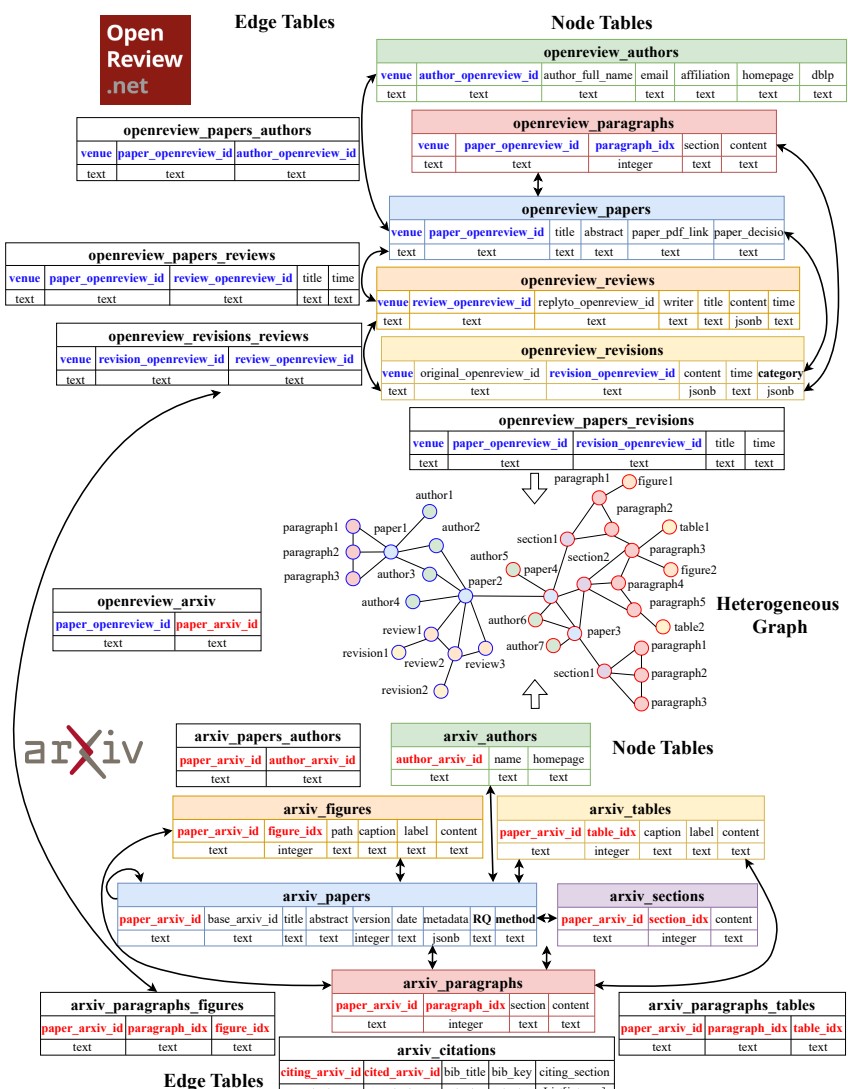

Figure 3: **A comprehensive overview of RESEARCHARCADE.** RESEARCHARCADE uses a multi-table format with graph structures to collect data from different sources with multiple modalities. Tables are classified into node tables (colored) or edge tables (black and white). The blue (denoting the OpenReview part) or red (denoting the ArXiv part) columns represent the unique identification of each node or edge, and the remaining columns represent the features of the nodes or edges. And the black bold columns are generated by LLM. The conversion from the multiple tables to heterogeneous graphs is straightforward.

**Stage 2: Author Information Processing**   Author names are first obtained from ArXiv metadata, which provides plain-text names without persistent identifiers. To enrich this information, we query the Semantic Scholar API using the ArXiv ID. We extract author identifiers, names, and optional auxiliary information (e.g., homepages). Authors are stored in a dedicated table, and paper–author relationships are recorded with sequence numbers to preserve author order.

**Stage 3: Section-Level Decomposition**   We identify paper sections by recursively traversing the LaTeX structure. The system assumes a three-level hierarchy (section, subsection, subsubsection). When sectioning commands (e.g., \section..., \subsection..., \subsubsection...) are detected, we extract section titles, store section content, and record their positions within the paper.

Table 5: **Statistic overview of the data collected from ArXiv by primary category.** Note that some papers may belong to multiple subdomains within a category. Counts are aggregated by top-level category prefix.

| Category | #papers | #sections | #paragraphs | #figures | #tables | #authors |
|---|---|---|---|---|---|---|
| cs | 57357 | 849448 | 11306247 | 1309302 | 528055 | 14385 |
| stat | 9938 | 106751 | 1879176 | 177750 | 47557 | 613 |
| physics | 8421 | 69334 | 1013456 | 122287 | 18395 | 349 |
| math | 5263 | 50397 | 1293357 | 71226 | 14666 | 766 |
| eess | 5069 | 37376 | 479536 | 54377 | 19070 | 1511 |
| q-bio | 2797 | 24588 | 353634 | 34342 | 10722 | 220 |
| cond-mat | 1991 | 16794 | 256023 | 22744 | 2905 | 81 |
| astro-ph | 690 | 6858 | 97977 | 10352 | 1962 | 47 |
| nlin | 388 | 3009 | 49485 | 4559 | 429 | 17 |
| econ | 300 | 2683 | 63133 | 3920 | 1371 | 30 |
| q-fin | 253 | 2468 | 42267 | 5360 | 1287 | 45 |
| **Total** | 66918 | 569501 | 8014095 | 876636 | 324648 | 14391 |

Table 6: **Statistic overview of the data collected from ICLR conferences, sourced from the OpenReview.** Note that no ICLR conference was held in 2015 and 2016. Additionally, revisions of submissions from the ICLR 2013 and 2014 conferences are not accessible on the OpenReview.

| Year | #papers | #authors | #reviews | #paragraphs | #revisions | #papers_authors | #papers_reviews | #papers_revisions | #reviews_revisions |
|---|---|---|---|---|---|---|---|---|---|
| 2025 | 8701 | 27742 | 190934 | 1526799 | 13989 | 42541 | 190934 | 13989 | 97051 |
| 2024 | 5750 | 18077 | 99525 | 389973 | 1251 | 25297 | 99520 | 1251 | 11971 |
| 2023 | 3793 | 11819 | 55301 | 893211 | 9445 | 15742 | 55301 | 9445 | 39871 |
| 2022 | 2617 | 8155 | 39750 | 614294 | 6508 | 10505 | 39750 | 6508 | 28321 |
| 2021 | 2594 | 7661 | 32113 | 566963 | 6593 | 9782 | 32113 | 6593 | 22786 |
| 2020 | 2213 | 6963 | 21132 | 556021 | 6878 | 9117 | 21132 | 6878 | 14773 |
| 2019 | 1419 | 4387 | 16620 | 306915 | 3671 | 5618 | 16620 | 3671 | 11503 |
| 2018 | 935 | 2820 | 9164 | 352761 | 4929 | 3512 | 9164 | 4929 | 8374 |
| 2017 | 490 | 606 | 6988 | 104648 | 1203 | 869 | 6988 | 1203 | 4206 |
| 2014 | 69 | 65 | 548 | 2803 | / | 84 | 548 | / | / |
| 2013 | 67 | 56 | 373 | 2691 | / | 74 | 373 | / | / |
| **Total** | 28648 | 88351 | 472448 | 5317079 | 54467 | 123141 | 472443 | 54467 | 238856 |

**Stage 4: Paragraph-Level Information Extraction** Within each section, text is segmented into paragraphs based on blank lines, explicit line breaks, and environment boundaries. Content belonging to figures, tables, or display-math environments is excluded to preserve clean textual paragraphs. Each paragraph is assigned paper-level and section-level ordering. We detect citation commands (e.g., `\cite...`) and extract citation keys and titles. References to figures and tables are identified through cross-reference commands (e.g., `\ref...`). These links are stored in relational tables that connect paragraphs to citations, figures, and tables.

**Stage 5: Citation Information Processing** Citation metadata is collected from BibTeX files and compiled bibliography environments (e.g., `.bbl` files or `\begin{thebibliography}` blocks). We extract citation keys, titles, authors, and venue information, and detect ArXiv identifiers when present. For references without explicit ArXiv IDs, we apply a two-step resolution process. First, we query Semantic Scholar to retrieve reference lists with external identifiers. Second, we align these references with local entries using normalized title similarity with a conservative threshold. Unmatched references are retained in the database without ArXiv IDs.

**Stage 6: Figure and Table Extraction** Figures and tables are detected and indexed using their environment labels and captions at both document and section levels. Each figure or table is stored as a structured object linked to its parent paper and, when applicable, its enclosing section. At the paragraph level, cross-references to figures and tables are resolved using a global label index. Only figures and tables that are referenced in the text are retained, ensuring semantic grounding. This enables fine-grained links between paragraphs and the visual or tabular content they describe.

Table 7: **Statistic overview of the data collected from NeurIPS conferences, sourced from the OpenReview.** Note that no NeurIPS conference data before 2021 is available on OpenReview. Additionally, no revision is allowed during the rebuttal process.

| Year | #papers | #authors | #reviews | #paragraphs | #papers_authors | #papers_reviews |
|------|---------|----------|----------|-------------|-----------------|-----------------|
| 2025 | 5529 | 21687 | 125739 | 370664 | 30352 | 125739 |
| 2024 | 4236 | 15430 | 75588 | 279708 | 21673 | 75588 |
| 2023 | 3394 | 11191 | 64528 | 218856 | 15863 | 64528 |
| 2022 | 2824 | 9102 | 46915 | 180202 | 12561 | 46915 |
| 2021 | 2768 | 7600 | 37952 | 167213 | 11744 | 37952 |
| **Total** | 18751 | 65010 | 350722 | 1216643 | 92193 | 350722 |

Table 8: **Statistic overview of the data collected from ICML conferences, sourced from the OpenReview.** Note that no ICML conference data before 2023 is available on OpenReview. Additionally, peer review data is only available for the year 2025, no revision is allowed during the rebuttal process, and no extracted paragraph data because the ICML PDFs are double-column-structured.

| Year | #papers | #authors | #reviews | #papers_authors | #papers_reviews |
|------|---------|----------|----------|-----------------|-----------------|
| 2025 | 3422 | 13279 | 38974 | 17871 | 38974 |
| 2024 | 2610 | 9516 | / | 13050 | / |
| 2023 | 1828 | 6186 | / | 8121 | / |
| **Total** | 7860 | 28981 | 38974 | 39042 | 38974 |

**Stage 7: Graph Construction and Storage**   All extracted elements are organized into a heterogeneous graph. Nodes represent papers, sections, paragraphs, figures, tables, and authors. Edges encode relationships such as paper–section, section–paragraph, paragraph–citation, paragraph–figure, paragraph–table, paper–author, paper–paper citations, and paper–category links. Citation edges are deduplicated and store both original bibliographic metadata and resolved ArXiv identifiers. This graph serves as the foundation for downstream tasks.

### A.2.2 CONTINUOUS CRAWLING FROM ARXIV

RESEARCHARCADE supports continuous and automated data acquisition through a fault-tolerant crawling and processing pipeline. The pipeline design reflects the multi-stage extraction procedure described above and enables the dataset to remain synchronized with newly released ArXiv papers while preserving consistency of the existing heterogeneous graph.

The major stages of the pipeline are as follows:

1. **ArXiv Identifier Retrieval.** A scheduled job (e.g., daily or weekly) queries the ArXiv API to collect newly published paper identifiers within a predefined time window, supporting both incremental updates and recovery from transient failures.

2. **Source Archive Downloading.** For each new ArXiv ID, the compressed LaTex source archive is automatically downloaded and unpacked into a staging directory to support parsing and processing.

3. **Incremental Graph Construction and Update.** Each paper is processed using the same pipeline described in Appendix A.2.1, converting source files into structured graph entities, including papers, sections, paragraphs, figures, tables, citations, and their relations.

4. **External Metadata Enrichment.** Additional metadata are added using external services, including author enrichment via the Semantic Scholar API and citation resolution through title and ArXiv ID based matching.

This automated pipeline allows RESEARCHARCADE to evolve continuously by integrating newly published content in a consistent and reproducible way, supporting longitudinal analysis of academic structures and relationships.

Table 9: **Statistic overview of the data collected from EMNLP conferences, sourced from the OpenReview.** Note that only the EMNLP conference in 2023 is available on OpenReview. Additionally, no revision is allowed during the rebuttal process.

| Year | #papers | #authors | #reviews | #paragraphs | #papers_authors | #papers_reviews |
|------|---------|----------|----------|-------------|-----------------|-----------------|
| **2023** | 2019 | 6696 | 22731 | 118184 | 10015 | 22731 |

Table 10: **Statistic overview of** openreview_arxiv **table on ICLR.** Note that no ICLR conference was held in 2015 and 2016. Additionally, revisions of submissions from the ICLR 2013 and 2014 conferences are not accessible on the OpenReview.

| Year | 2025 | 2024 | 2023 | 2022 | 2021 | 2020 | 2019 | 2018 | 2017 | 2014 | 2013 |
|------|------|------|------|------|------|------|------|------|------|------|------|
| **#openreview_arxiv** | 3077 | 2033 | 1469 | 1050 | 1068 | 866 | 583 | 424 | 248 | 53 | 50 |

The pipeline also supports two interchangeable storage backends: a PostgreSQL database and a CSV-based file store. PostgreSQL provides efficient querying and scalability for large-scale experiments, while the CSV backend offers a lightweight, portable option that requires no database deployment and can be easily used across different computing environments.

### A.2.3 OpenReview

The detailed procedures used to collect and compile data from OpenReview are as follows.

Firstly, by providing a conference ID, we utilize the OpenReview API to retrieve the authors' IDs, titles, abstracts, decisions, PDF links, and unique submission IDs for each paper presented at the conference. Note that we do not collect the withdrawn papers. This step mainly contributes to the construction of the or_papers table and the or_papers_authors table.

Given the author IDs, the OpenReview API returns detailed author metadata, including full name, email domain, institutional affiliation, homepage URL, and DBLP entry. Note that, for some authors, the homepage and DBLP fields are missing from the metadata. These records constitute the authors table. Moreover, each author in OpenReview has a unique author ID, although they might have the same name.

The OpenReview API also provides access to official reviews and comments associated with each paper submission. For each review, we retrieve its ID, the ID of the review it responds to, and its timestamp. It is important to note that the official review, meta-review, and paper decision directly reply to the submission ID. The collected data is then used to form the or_reviews table and the or_papers_reviews table.

To construct the or_paragraphs table, we first download the PDF files and utilize pdfminer to extract the text from papers. The extracted text is then organized into paragraphs with unique paragraph_idx within each paper based on the distance between consecutive words.

For the or_revisions table and the or_papers_revisions table, we focus on the content of the revisions. Therefore, we begin by downloading the PDFs of all the revised papers and storing their content in the or_paragraphs table. Note that each revised paper also has a unique submission ID. Then, based on the timestamp of each revised submission, we identify pairs of original and revised papers and use difflib to extract the differences between the original and the revised texts. Finally, the locations of these differences are linked back to each paragraph.

Finally, to construct the or_revisions_reviews table, we assume that each revision is created as a result of discussions between the reviewers and the authors. And the specific discussions are identified if they occurred within the time period between the previous revision and the current revision. Thus, this table is constructed by leveraging the time information from the or_revisions table and or_reviews table.

Table 11: **Statistic overview of** openreview_arxiv **table on NeurIPS.** Note that no NeurIPS conference before 2021 is available on OpenReview.

| Year | 2025 | 2024 | 2023 | 2022 | 2021 |
|---|---|---|---|---|---|
| **#openreview_arxiv** | 2898 | 2303 | 1808 | 1430 | 1368 |

Table 12: **Statistic overview of** openreview_arxiv **table on ICML.** Note that no ICML conference before 2023 is available on OpenReview.

| Year | 2025 | 2024 | 2023 |
|---|---|---|---|
| **#openreview_arxiv** | 1828 | 1398 | 998 |

#### A.2.4 LLM-GENERATED CONTENT

To facilitate more diverse tasks, we further preprocess the collected data using Llama-3.1-70B-Instruct (Grattafiori et al., 2024). Specifically, we classify the entities for each revision in the or_revisions table, and summarize the research question and method for each academic paper in the ar_papers table.

**Revision Classification**: We follow the previous work (Jourdan et al., 2025) to classify the revisions into 9 categories described in Table 14. And the specific prompt is listed in Table 15.

To validate whether the model is capable of classifying the revision, we test three state-of-the-art LLMs, Llama-3.1-70B-Instruct (Grattafiori et al., 2024), GPTOSS-120B (Agarwal et al., 2025), and GPT-4o-mini (Hurst et al., 2024) on the revision dataset Pararev from Jourdan et al. (2025). It contains 641 human-annotated examples. The evaluation results are shown in Table 18. Since Llama-3.1-70B achieves the best performance, we use it to classify the revision to the best of our knowledge.

**Research Question and Method Summarization**: Similarly, in order to facilitate tasks like scientific claim verification, document-level retrieval, and hypothesis synthesis, we preprocess the data by generating structured research questions and method descriptions for each paper. The specific prompts are shown in Table 16 and Table 17. These LLM-generated summaries provide a normalized, compact representation of a paper's core intent and technical approach, enabling more reliable cross-paper comparison, semantic indexing, and graph-based reasoning in downstream tasks. By transforming unstructured section text into standardized question–method pairs, we improve both the efficiency and interpretability of higher-level analytical modules built on top of RESEARCHAR-CADE.

### A.3 EVALUATION TASK DEFINITIONS

#### A.3.1 CITATION PREDICTION

**Step 1**: The target entity $t$ is an arxiv_paragraph_citation edge, representing a citation made by a source paragraph. The label $\mathbf{y}_t$ denotes the index of the ground-truth cited paper that the target paragraph should reference.

**Step 2**: The academic graph $\mathcal{G}_t$ corresponds to the full paper containing the target paragraph. It includes arxiv_section, arxiv_paragraph, arxiv_figure, and arxiv_table nodes. Paragraphs are sequentially linked. arxiv_section nodes provide the section title; arxiv_citation nodes represent external cited papers and are connected to citing paragraphs via arxiv_paragraph_citation edges.

#### A.3.2 PARAGRAPH GENERATION

**Step 1**: The target entity $t$ is an arxiv_paragraph node, with its textual content serving as the ground truth label $\mathbf{y}_t$.

**Step 2**: The academic graph $\mathcal{G}_t$ for this task includes the adjacent arxiv_paragraph nodes retrieved from the $k$-hop neighborhood (with $k$ as a parameter), sequentially connected according to their

Table 13: **Statistic overview of** openreview_arxiv **table on EMNLP.** Note that only the EMNLP conference in 2023 is available on OpenReview.

| Year | 2023 |
|---|---|
| **#openreview_arxiv** | 1017 |

Table 14: **Category description of** category **column in** openreview_revisions **table.** It follows the category description in Jourdan et al. (2025).

| Category | Description |
|---|---|
| Rewriting Light | Minor changes in word choice or phrasing. |
| Rewriting Medium | Complete rephrasing of sentences within the paragraph. |
| Rewriting Heavy | Significant rephrasing, affecting at least half of the paragraph. |
| Concision | Same idea, stated more briefly by removing unnecessary details. |
| Development | Same idea, expanded with additional details or definitions. |
| Content Addition | Modification of content through the addition of a new idea. |
| Content Substitution | Modification of content through the replacement of an idea or fact. |
| Content Deletion | Modification of content through the deletion of an idea. |
| Unusable | Issues due to document processing errors (e.g., segmentation problems, misaligned paragraphs, or footnotes mixed with the text). |

order in the paper. Multi-modal nodes arxiv_figure and arxiv_table are also given, each linked to their corresponding paragraphs. arxiv_citation is added as external nodes connected to the citing paragraphs.

### A.3.3 Revision Retrieval

**Step 1**: The target entity $t$ in this task is an openreview_revision node, where the index list of the modified paragraphs in its attributes is the label $\mathbf{y}_t$ for this task.

**Step 2**: The academic graph $\mathcal{G}_t$ constructed in this task consists of two parts: First, the paragraphs from the original paper, with node type openreview_paragraph, are retrieved from the 2-hop neighborhood, according to the openreview_paper_revision and the openreview_paragraph table. These paragraphs are sequentially connected based on their order; Second, the reviews, with node type openreview_review, are also retrieved from the 2-hop neighborhood, according to the openreview_paper_revision and the openreview_papers_review table. They are connected based on their review_openreview_id and replyto_openreview_id attributes.

### A.3.4 Revision Generation

**Step 1**: The target entity $t$ in this task is a paragraph that has been revised. A revised paragraph is obtained based on the revision_openreview_id and the index list of the modified paragraphs for each openreview_revision node. The textual content of the revised paragraph is the label $\mathbf{y}_t$.

**Step 2**: To construct the academic graph $\mathcal{G}_t$ for this task, two types of nodes from $t$'s neighborhood need to be retrieved: First, the corresponding paragraph from the original paper, with node type openreview_paragraph, is retrieved from the 2-hop neighborhood based on the corresponding openreview_revision node and the openreview_paragraph table; Second, the reviews, with node type openreview_review, are also retrieved from the 2-hop neighborhood based on the corresponding openreview_revision node, along with the openreview_paper_revision and the openreview_papers_review tables. These reviews are connected via their review_openreview_id and replyto_openreview_id attributes.

### A.3.5 Acceptance Prediction

**Step 1**: Node or_paper is the target entity $t$ in this task, and the paper's decision (Accept or Reject) is the label $\mathbf{y}_t$.

| Role | Content |
|------|---------|
| System | You are an experienced academic researcher. You will receive an original paragraph and its corresponding revised paragraph. Your task is to analyze the revision and determine its taxonomy. The revision can receive at most two taxonomies. 

 The description of each taxonomy is as follows. 
 [Rewriting Light]: Minor changes in word choice or phrasing. 
 [Rewriting Medium]: Complete rephrasing of sentences within the paragraph. 
 [Rewriting Heavy]: Significant rephrasing, affecting at least half of the paragraph. 
 [Concision]: Same idea, stated more briefly by removing unnecessary details. 
 [Development]: Same idea, expanded with additional details or definitions. 
 [Content Addition]: Modification of content through the addition of a new idea. 
 [Content Substitution]: Modification of content through the replacement of an idea or fact. 
 [Content Deletion]: Modification of content through the deletion of an idea. 
 [Unusable]: Issues due to document processing errors (e.g., segmentation problems, misaligned paragraphs, or footnotes mixed with the text). 

 Please give concrete evidence while being concise. DO NOT repeat or summarize the revision's content or similarities; focus on their differences and YOUR ANALYSIS. 
 Output [START]{{[Taxonomy]}}[END] or [START]{{[Taxonomy 1], [Taxonomy 2]}}[END] |
| User | <Original Paragraph>: {original_paragraph}, 
 <Revised Paragraph>: {revised_paragraph} |

Table 15: **Prompt for Revision Classification.**

| Role | Content |
|------|---------|
| User | Based on the following sections from a research paper, identify and summarize the main research question(s) or objective(s). 

 Your summary should: 
 - Be concise (2-4 sentences) 
 - Clearly state what problem the paper addresses 
 - Mention the key contributions or goals 

 Do not include methodology details. 

 Paper Sections: 
 {section_text} 

 Summary: |

Table 16: **Prompt for Paper Research Question Generation.**

**Step 2**: The academic graph $\mathcal{G}_t$ is constructed using the data from ArXiv: First, relevant paragraphs, with node type arxiv_paragraph, are retrieved from the 2-hop neighborhood, according to the openreview_arxiv and the arxiv_paragraph tables, with sequential connections reflecting their order. Second, the related figures, with node type arxiv_figure, are retrieved through the arxiv_paragraph_figure table, with each figure connected to a specific paragraph. Finally, relevant tables, with node type arxiv_table, are retrieved via the arxiv_paragraph_table table.

### A.3.6 Rebuttal Generation

**Step 1**: The author's rebuttal response (can be inferred from the openreview_review node's title), with node type openreview_review, is the target entity $t$ in this task. The label $\mathbf{y}_t$ is the textual content of the response.

**Step 2**: The academic graph $\mathcal{G}_t$ is constructed as follows: Initially, the related official review, with node type openreview_review, is retrieved based on the replyto_openreview_id attribute of $t$. Then, the corresponding paper graph is retrieved from ArXiv data using the same procedure as in Section 4.2.5, which contains the relevant paragraphs with figures and tables.

### A.4 Promising New Tasks

In this part, we list out and describe what tasks can be performed on RESEARCHARCADE in each research stage.

| Role | Content |
|------|---------|
| User | Based on the following sections from a research paper, summarize the main methodology or approach used.

Your summary should:
- Be concise (3-5 sentences)
- Describe the key techniques, algorithms, or frameworks
- Mention any novel components or modifications
- Briefly note the experimental setup if relevant
Do not include research questions or results.

Paper Sections:
{section_text}

Summary: |

Table 17: **Prompt for Paper Method Description Generation.**

Table 18: **Evaluation results on Revision Classification.**

| Model | Accuracy |
|-------|----------|
| Llama3.1-70B-Instruct | **0.541** |
| GPTOSS-120B | 0.468 |
| GPT-4o-mini | 0.512 |

### A.4.1 IDEA GENERATION

Brainstorming research ideas based on existing works is an essential skill for any researcher. Enhancing the model's ability to support this task facilitates the idea brainstorming stage in the research pipeline. This generative task is defined as follows: given the input, an academic graph $\mathcal{G}_t$ containing the abstract of the papers that are being cited, generate the abstract of the citing paper $\hat{\mathbf{y}}_t$. The label $\mathbf{y}_t$ is the real abstract of the paper.

### A.4.2 EXPERIMENT PLANNING

Planning an experiment to verify the effectiveness of the work is a necessary part of doing research. This generative task is defined as follows: given the input, an academic graph $\mathcal{G}_t$ consisting of paragraphs with figures and tables before the experiment section, generate the main experiment table text $\hat{\mathbf{y}}_t$. The real experiment table text is the label $\mathbf{y}_t$.

### A.4.3 ABSTRACT WRITING

Writing a high-quality abstract is a challenging but meaningful task. This generative task is defined as follows: given the input, an academic graph $\mathcal{G}_t$ including all paragraphs with figures and tables from the paper, generate its abstract $\hat{\mathbf{y}}_t$. The label $\mathbf{y}_t$ is the real abstract.

### A.4.4 REVIEW GENERATION

Automatic generation of reviews can serve as a paper copilot, aiding the improvement of the manuscript. The task reflects the peer reviewing stage in the research pipeline. This generative task is defined as follows: given the input, an academic graph $\mathcal{G}_t$ containing paragraphs of the paper, generate its official review $\hat{\mathbf{y}}_t$. The real official review is the label $\mathbf{y}_t$.

## A.5 DATA USAGE IN EXPERIMENTS

### A.5.1 CITATION PREDICTION

In this task, we use 1,267 ar_paper nodes containing 41,31 sampled ar_citation edges. Each paper also includes ar_section nodes and ar_citation edges. The ar_citation edges are split into 33,048 for training and 827 for testing.

Table 19: **LLM-as-a-judge Prompt for Rebuttal Generation.**

| Role | Content |
|---|---|
| System | You are an experienced academic paper reviewer. You will receive a review of an academic paper in computer science, and two responses from the authors. (Response 1 & Response 2) 
 Your task is to evaluate the responses and decide which response is better. 

 The response may address the reviewer's several comments. You should compare the responses to each comment individually. 
 When comparing the responses, you can refer to the following criteria: 
 - 1. Does the author's response validate their work with clear arguments and coherent logic? 
 - 2. Does the author provide sufficient evidence or reasoning to support their claims? 
 - 3. Is the author's response consistent with the content of the original paper? 
 Please give concrete evidence while being concise. DO NOT repeat or summarize the responses' content or similarities; focus on their differences and YOUR ANALYSIS. 
 Output [START]{{I think Response X (1 or 2) is better}}[END] or [START]{{I think Response 1 and Response 2 are similar in quality}}[END] |
| User | [Review]: {official_review} 
 [Response 1]: {target} 
 [Response 2]: {predic} |

### A.5.2 PARAGRAPH GENERATION

In this task, we use 1,200 ar_paragraph nodes together with their connected ar_figure, ar_table, ar_citation nodes and edges, and adjacent ar_paragraph nodes.

### A.5.3 REVISION RETRIEVAL

The set of target entities for this task comprises 5,000 or_revision nodes from ICLR 2025, split into 4,000 for training and 1,000 for testing.

### A.5.4 REVISION GENERATION

Using 5,000 or_revision nodes from ICLR 2025—split 4,000/1,000 into train/test—yields 27,892 and 8,821 revised paragraphs, with node type or_paragraph for training and testing, respectively.

### A.5.5 ACCEPTANCE PREDICTION

In this task, the test set comprises 300 or_paper nodes from ICLR 2025 that are linked to an ar_paper via the or_ArXiv table. The training set contains 1,200 nodes—300 each from ICLR 2021–2024—selected under the same linkage criterion.

### A.5.6 REBUTTAL GENERATION

We select 3,077 and 2,898 or_paper nodes and their corresponding official reviews and author rebuttals from ICLR 2025 and NeurIPS 2025, respectively, each or_paper node is connected to an ar_paper node via the openreview_arxiv table. For the training and test sets, we split them into 2,779/298 and 2,680/290 papers.

## A.6 PROMPT USAGE

### A.6.1 LLM-AS-A-JUDGE

The prompt for LLM-as-a-judge in rebuttal generation is in Table 19.

The prompt for LLM-as-a-judge in revision generation is in Table 20.

The prompt for LLM-as-a-judge in paragraph generation is in Table 21.

### A.6.2 PARAGRAPH GENERATION

The prompt used for the GWM-based models in the paragraph generation task is in Table 22. Specifically, {title}, {abstract}, {title}, {section name}, {figure labels and captions}, {citation bib}, {title} are text-based tokens, where {paper_graph} is the embedding-based tokens that are processed by multi-hop aggregation.

Table 20: **LLM-as-a-judge Prompt for Revision Generation.**

| Role | Content |
|---|---|
| System | You are an experienced academic peer reviewer. You will receive reviews of an academic paper in computer science, a paragraph in the original paper, and two revised paragraphs of the original paragraph. (Revision 1 & Revision 2) 
 Your task is to evaluate the revisions and decide which revision is better. 

 The revision may address the reviewer's several comments. You should compare the two revisions to each comment individually. 
 When comparing the revisions, you can refer to the following criteria: 
 - 1. Does the author's revision solve the problem stated in the reviews? 
 - 2. Does the author's revision validate their work with clear arguments and coherent logic? 
 - 3. Is the author's revision consistent with the content of the original paper? 
 Please give concrete evidence while being concise. DO NOT repeat or summarize the responses' content or similarities; focus on their differences and YOUR ANALYSIS. 
 Output [START]{{I think Revision X (1 or 2) is better}}[END] or [START]{{I think Revision 1 and Revision 2 are similar in quality}}[END] |
| User | [Reviews]: {reviews} 
 [Original Paragraph]: {original_paragraph} 
 [Revision 1]: {target} 
 [Revision 2]: {predic} |

Table 21: **LLM-as-a-judge Prompt for Paragraph Generation.**

| Role | Content |
|---|---|
| System | You are an experienced academic peer reviewer. You will receive: 
 (1) the surrounding context of an academic paper, 
 (2) the ground-truth paragraph that was originally masked out, and 
 (3) four model-generated regenerated paragraphs (Revision 1, Revision 2, Revision 3, Revision 4). 
 Your task is to evaluate the four regenerated paragraphs and decide which one is the best reconstruction of the masked paragraph. 
 You should compare the revisions against the ground-truth paragraph and the surrounding paper context. 
 When comparing the revisions, use the following criteria: 
 - 1. Fidelity: How accurately does the revision preserve the meaning, technical content, and intent of the ground-truth paragraph? 
 - 2. Coherence: How well does the revision fit logically and stylistically with the surrounding paper context? 
 - 3. Clarity: How clear, precise, and academically appropriate is the writing? 
 - 4. Completeness: Does the revision capture all key points without adding unsupported information? 
 Provide concrete evidence while being concise. DO NOT repeat or summarize the revision contents; focus only on their differences and YOUR ANALYSIS. 
 Output format (strict): 
 [START]{I think Revision X (1, 2, 3, or 4) is better}[END] 
 or 
 [START]{I think the revisions are similar in quality}[END] |
| User | Paper Context: 
 {context} 
 Ground-Truth Paragraph: 
 {ground_truth} 
 Revision 1 (hop0): 
 {generated_paragraph[0]} 
 Revision 2 (hop1): 
 {generated_paragraph[1]} 
 Revision 3 (hop3): 
 {generated_paragraph[2]} 
 Revision 4 (hop5): 
 {generated_paragraph[3]} 
 Please evaluate which revision best reconstructs the ground-truth paragraph. |

### A.6.3 REVISION RETRIEVAL

The prompt used for the GWM-based models in the revision retrieval task is in Table 23. Specifically, {review_graph} is an embedding-based token that is processed by multi-hop aggregation.

### A.6.4 REVISION GENERATION

The prompt used to let LLM summarize the review is in Table 24. Specifically, {review} is the text-based content of a single review.

The following prompt used in the rebuttal generation task is in Table 25. Specifically, {review_graph} is the text-based token that sequentially connects the reviews. (e.g., Official Review by Reviewer, ...; Response by Authors: ...)

### A.6.5 ACCEPTANCE PREDICTION

The prompt used for the GWM-based models in the acceptance prediction task is in Table 26. Specifically, {paper_graph} is an embedding-based token that is processed by multi-hop aggregation.

Table 22: **Prompt for Paragraph Generation.**

| Role | Content |
|------|---------|
| User | {paper_graph} You are reconstructing one missing LaTeX paragraph in a research paper.
Title: {title}
Abstract: {abstract}
Section: {section name}
Figure (optional): {figure labels and captions}
Table (optional): {table labels and captions}
Citation (optional): {citation bib}
Generate the missing paragraph between the next paragraphs and previous paragraphs in the embedding space; feel free to use the given figure, table and citation information. |

Table 23: **Prompt for Revision Retrieval.**

| Role | Content |
|------|---------|
| User | {review_graph}. Analyze the rebuttal process between the reviewer and the authors to identify information suggesting necessary modifications to the paper. |

### A.6.6 REBUTTAL GENERATION

The prompt used in the rebuttal generation task is in Table 27. Specifically, {paper_graph} are text-based tokens that sequentially link paragraphs, while figures and tables explicitly denote their connections to the paragraphs (e.g., Paragraph 1: {paragraph content}, Figure: {figure description}, Table: {table text}; Paragraph 2: ...).

### A.6.7 ABLATION STUDY ON REVIEW INFORMATION

The prompt used in the ablation study on review information is in Table 28.

### A.7 EXPERIMENT RESULTS ANALYSIS

### A.7.1 CITATION PREDICTION

In this subsection, we present the evaluation results for citation prediction.

**Baselines.** We compare embedding-based models with GNN-based models using 1-, 3-, and 5-hop neighborhood aggregation.

**Experimental Results.** Table 3 reveals two key findings: (1) incorporating adjacent neighborhoods provides sufficient contextual information and substantially improves prediction accuracy; and (2) expanding the receptive field to larger neighborhoods yields additional performance gains, although these improvements stop when increasing from 3-hop to 5-hop neighborhoods. This trend is likely due to the sparsity of the paper graph and the incomplete coverage of citation, figure, and table information during data processing, which limits the benefits of incorporating more distant nodes.

### A.7.2 PARAGRAPH GENERATION

In this subsection, we present the evaluation results for generating missing paragraphs.

**Ablation Study** We conducted two types of ablation studies for this task. The first evaluates multi-hop paragraph generation by varying the amount of neighborhood information provided across different hops. The second test involved multi-modal inputs using four conditions: both figures and tables, figures only, tables only, and neither component.

**Experiment Results** Table 3 reveals two main findings for neighborhood information: (1) models achieve their best performance when given the most comprehensive neighborhood context, and (2) removing neighborhood information leads to substantial performance degradation.

Table 24: **Prompt for Review Summarization.**

| Role | Content |
|------|---------|
| User | REVIEW: {review} 

 INSTRUCTIONS: 
 - Summarize the following review into fewer than 150 words. 
 - Output only the summarization, enclosed between [START] and [END], without any extra explanation or analysis. 

 OUTPUT: [START]{{your summarization here}}[END] |

Table 25: **Prompt for Revision Generation.**

| Role | Content |
|------|---------|
| User | REVIEW: {review_graph} 

 ORIGINAL PARAGRAPH: {original_paragraph} 

 INSTRUCTIONS: 
 - Please revise the paragraph according to the provided reviews. 
 - Output only the revised paragraph, enclosed between [START] and [END], without any extra explanation or analysis. 

 REVISED PARAGRAPH: [START]{{your revised paragraph here}}[END] |

For multi-modal information, the results show that (1) complete multi-modal input (both figures and tables) yields the strongest performance, and (2) partial multi-modal input offers no clear benefit over omitting multi-modal data entirely.

Human analysis further indicates that paragraphs generated with larger-hop neighborhoods tend to contain more specific details (e.g., correctly named methods, datasets, and figure/table references) and exhibit smoother connections to adjacent paragraphs. This makes the generated text better integrated into the full paper rather than resembling a stand-alone summary. At 5 hops, however, the text often becomes more verbose and less focused, with repeated points, additional concepts, and overly detailed explanations, so although it is richer than 0- or 1-hop outputs, it also exhibits more topical drift and redundancy.

GPT-4o-mini, used as an LLM-as-a-judge, shows a clear preference for models with richer graph context: its scores increase from 0-hop (0.009) to 1-hop (0.244) and 3-hop (0.404), indicating that multi-hop information yields outputs it rates as higher quality. However, the score decreases at 5-hop (0.344), suggesting that expanding the neighborhood beyond 3 hops does not provide additional gains in judged quality and may slightly reduce it.

### A.7.3 REVISION RETRIEVAL

**Baselines.** Embedding-based, GNN-based, and GWM-based models are selected as our baselines. Specifically, we consider 1-hop and 3-hop aggregation for GNN-based and GWM-based models.

**Experimental Results.** From Table 3 we can observe that: (1) Models optimized with InfoNCE (GNN-/GWM-based) outperform the untrained embedding baseline, confirming the effectiveness of our training and the quality of RESEARCHARCADE. (2) Graph-aware models consistently exceed non-graph baselines (EMB-based), indicating that relational structure provides a valuable signal for the task. (3) Increasing the message-passing radius yields little to no additional gain; we attribute this to the sparsity and near-sequential topology of review-centered graphs for most samples, which limits the benefits of multi-hop aggregation and may introduce noise or over-smoothing.

### A.7.4 REVISION GENERATION

**Baselines.** For the generative task, we use LLM-based models as baselines. Qwen3-8B and GPTOSS-120B are evaluated in a zero-shot setting, while Qwen3-0.6B is evaluated under both zero-shot and supervised fine-tuning (SFT) settings.

Table 26: **Prompt for Acceptance Prediction.**

| Role | Content |
|------|---------|
| User | {paper_graph}. Analyze whether this academic paper is suitable for acceptance at the ICLR conference. |

Table 27: **Prompt for Rebuttal Generation.**

| Role | Content |
|------|---------|
| User | REFERENCES: {paper_graph}

QUESTIONS: {official_review}

INSTRUCTIONS:
- You are the author responding to the reviewer's comments during the rebuttal process.
- Generate the author's response based on the provided references from the paper (include paragraphs, figures, and tables).
- Provide ONLY the final response enclosed between [START] and [END], without any additional explanation or analysis.

Generated Rebuttal: [START]{{author's response here}}[END] |

**Experimental Results.** The results are displayed in Table 3, with the following observations: (1) There are substantial performance gaps between zero-shot Qwen3-0.6B and Qwen3-8B or GPTOSS-120B, which are reasonable in view of their different sizes of parameters. Human analysis indicates that larger models provide more relevant technical details based on their own knowledge than smaller models, thereby achieving higher performance. (2) After supervised fine-tuning Qwen3-0.6B, its performance is significantly enhanced, approaching the zero-shot performance of Qwen3-8B and GPTOSS-120B, highlighting the effectiveness of RESEARCHARCADE in facilitating LLMs' understanding of the dynamic evolution within a paper. (3) The GPTOSS-120B and Qwen3-8B obtain significantly high LLM-as-a-judge scores (Eq. 6). Human analysis of the examples suggests that compared with ground truth revisions, the LLM-generated revisions only make superficial changes to the original paragraphs. And GPT-4o-mini prefers the revised paragraphs that are similar to the original paragraphs. However, authors typically make more substantial edits to improve the paragraph, such as strengthening the logic by changing the narrative sequence or emphasizing the viewpoint by adding new information.

### A.7.5 ACCEPTANCE PREDICTION

**Baselines.** MLP-based, GNN-based, and GWM-based models are adopted as the baselines for this binary classification task. Here, 1-hop and 3-hop aggregation are considered for GNN-based and GWM-based models.

**Experimental Results.** As shown in Table 3, the results yield the following findings: (1) The best baseline achieves only $0.550$ accuracy, highlighting the challenge of predicting paper acceptance. (2) Graph-based models (GNN-based, GWM-based) outperform the non-graph-based model (MLP-based), which suggests that containing graph-structured data improves models' performance. This also confirms the validity of the highly relational and heterogeneous feature of RESEARCHAR-CADE. (3) The GNN-based model and GWM-based model with multi-hop aggregation achieve performance gain, indicating that multi-hop message passing further enhances the utilization of the graph-structured data.

### A.7.6 REBUTTAL GENERATION

**Baselines.** In the generative setting, LLM-based models are adopted as our baselines. Specifically, Qwen3-8B and GPTOSS-120B are assessed under a zero-shot manner, whereas Qwen3-0.6B is evaluated in both zero-shot and supervised fine-tuning (SFT) manners.

**Experimental Results.** The results in Table 3 reveal the following insights: (1) There exists substantial performance gaps between Qwen3-0.6B and Qwen3-8B or GPTOSS-120B in the zero-shot setting, which meets our expectations given their different parameter sizes. (2) After supervised fine-tuning, the Qwen3-0.6B shows enhanced performance, underscoring the efficacy of RESEARCHAR-CADE. (3) GPTOSS-120B achieves a significantly higher LLM-as-a-judge scores (Eq. 6) than other

Table 28: **Prompt for Revision Retrieval Ablation Study.**

| Role | Content |
| --- | --- |
| User | Please revise the paper for better quality. |

Figure 4: **Case Study on Revision Retrieval.**

baselines. According to human analysis of the examples, GPTOSS-120B tends to directly incorporate generated technical details and experimental results into its response, different from real authors, who usually refer reviewers back to specific portions of the paper. While other LLMs often generate shorter responses that only point to parts of the paper without further explanation, resulting in worse performance.

## A.8 CASE STUDY

We conduct several case studies across various models, including EMB-based, GNN-based, LLM-based, GWM-based models, and ChatPDF, a powerful AI-powered app proficient in comprehending PDFs.

### A.8.1 REVISION RETRIEVAL

In this section, we compare EMB-based, GNN-based, and GWM-based models on RESEARCHARCADE with ChatPDF for the Revision Retrieval task. The examples are shown in Figure 4.

In Figure 4, the comments from the reviewer focus on three aspects: (1) Standard deviations in Figure 2; (2) Ablation Study on SVD layers; (3) Comparison between SWIM-ODE and ELM-ODE. These correspond to four components in the paper: A. Figure 2; B. Section 3.2 (including

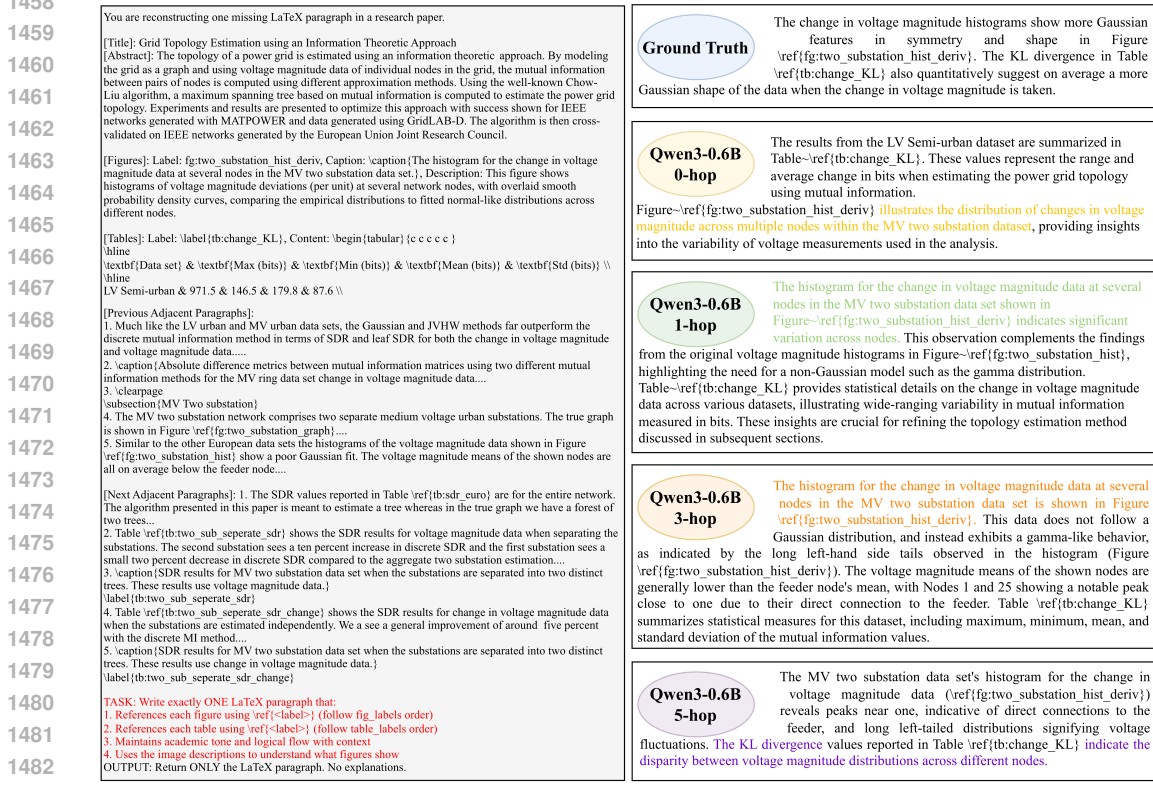

Figure 5: **Case Study on Paragraph Generation.**

the discussion on SWIM-ODE and ELM-ODE); C. Section 3.5 (including the discussion of the SVD-layers); D. Conclusion. The actual post-review revisions made by the authors are highlighted in light green (Note that the blue text is the overall revisions compared to the initial submission). Among them, although Figure 2 is explicitly mentioned by the reviewer, it remains unchanged. Most revisions, especially in Sections 3.2 and 3.5, add content directly addressing comments (2) and (3).

Examining the performance of different models, ChatPDF achieves the best results by correctly identifying three relevant components, demonstrating its superior ability to comprehend PDF files. The EMB-based model identifies only one relevant component, suggesting that it struggles with the mixed semantic information in the reviewer comments. For the GNN-based and GWM-based models, they exploit the graph structure within the paper and the rich semantic information at the embedding level to locate the two relevant components. Overall, while ChatPDF, as a state-of-the-art tool-augmented system, shows the strongest ability to understand PDF files, the graph modeling in RESEARCHARCADE is also valuable, as the graph-based models (GNN-based, GWM-based) outperform the non-graph-based model (EMB-based).

### A.8.2 PARAGRAPH GENERATION

The case study is shown in Figure 5. We analyze the generated paragraphs across different hop settings and observe a clear progression in how structural and semantic context is incorporated. The 0-hop answer primarily restates generic dataset-level information and provides weak grounding to the target figure and table, indicating limited contextual awareness. With 1-hop, the model begins to align the histogram in the figure with surrounding voltage magnitude discussions and introduces more meaningful connections to the KL statistics in the table, but still relies on high-level descriptions. The 3-hop setting demonstrates a substantial qualitative improvement: it correctly characterizes the non-Gaussian, gamma-like behavior of the distributions, links node-level behavior to feeder proximity, and integrates figure–table semantics in a logically coherent manner. At

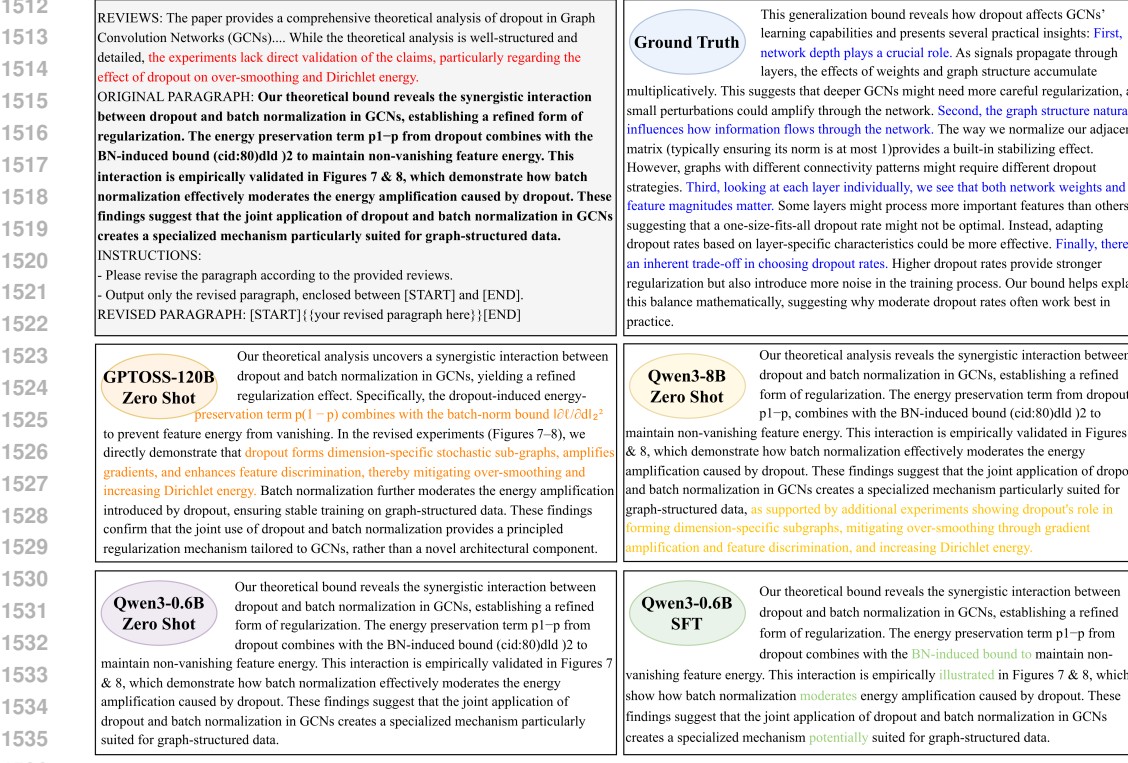

Figure 6: **Case Study on Revision Generation.**

5-hop, the model remains factually aligned but becomes more compressed and selective, focusing on a few salient patterns (peaks and tails) while losing some fine-grained statistical nuance. Overall, increasing hop depth enables the model to exploit broader graph context, moving from surface-level paraphrasing (0–1 hop) to structurally grounded, semantically rich reconstruction (3 hops), while very large hops (5) show a tendency toward abstraction and information smoothing.

### A.8.3    REVISION GENERATION

In this section, we compare different LLM-based models on RESEARCHARCADE for the Revision Retrieval task. The examples are shown in Figure 6.

In Figure 6, the comments from the reviewer point out the lack of direct validation on the effect of dropout on over-smoothing and Dirichlet energy. Instead of only making a high-level claim about the synergy between dropout and batch normalization, the actual revision explicitly unpacks the generalization bound into several concrete, testable insights to demonstrate the effect of dropout, providing a stronger statement with more details.

Examining the performance of different LLM-based models, they only make superficial changes to the original paragraph, limiting the effectiveness. To be more specific, zero-shot Qwen3-0.6B fails to make any changes to the original paragraph. Even after supervised fine-tuning, Qwen3-0.6B only deletes or replaces a few words, making no actual improvement. For Qwen3-8B and GPTOSS-120B in the zero-shot setting, they further introduce mentions of the dropout effect on over-smoothing and Dirichlet energy, but their improvements remain minimal compared with the authors' actual revision.

### A.8.4    REBUTTAL GENERATION

In this section, we compare LLM-based models on RESEARCHARCADE with ChatPDF for the Revision Retrieval task. The examples are shown in Figure 7.

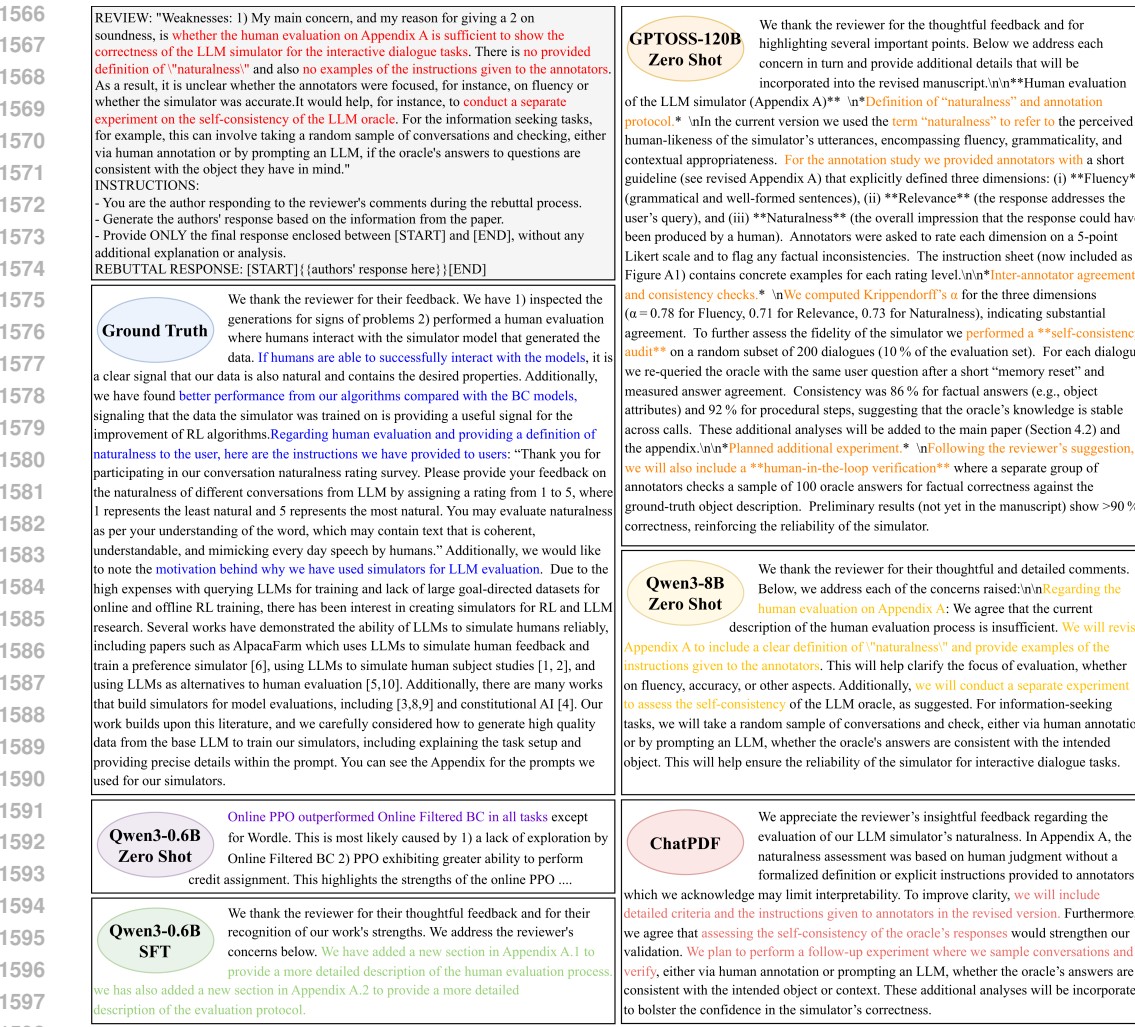

Figure 7: **Case Study on Rebuttal Generation.**

In Figure 7, the comments from the reviewer indicate that the correctness of the LLM simulator for the interactive dialogue is questionable. Specifically, 1) no definition of "naturalness" or examples of the instructions given to the annotators were provided; 2) a separate experiment on the self-consistency of LLM would be beneficial. The authors' response addresses the reviewer's concern in several ways. First, they argue that the successful interaction and better performance imply the correctness of the simulator. Second, they clarify the definition of "naturalness" and provide the actual instructions. Third, they also emphasize their motivation for using the LLM simulator with a broad literature.

Comparing the performance of different LLM-based models and ChatPDF, zero-shot GPTOSS-120B achieves the best performance: it covers all reviewer concerns and provides plausible technical details and experimental results in the response, although these are model-generated and may raise hallucination concerns. ChatPDF and zero-shot Qwen3-8B yield similar responses that address all aspects but lack details. Zero-shot Qwen3-0.6B fails to produce a meaningful answer and instead generates largely irrelevant content. After supervised fine-tuning, Qwen3-0.6B learns the general response pattern but still omits some aspects and lacks details. Overall, the ability of the larger LLM-based models to produce high-quality rebuttal responses verifies the efficacy of RE-SEARCHARCADE.

### A.9 LLM Usage Disclosure

We used large language models (LLMs) to assist with literature search and identification of related work relevant to our research on graph-based academic data interfaces. Specifically, we employed LLMs to help discover papers across different research areas that intersect with our work, including graph neural networks, large language models, academic data mining, and research automation. All identified papers were subsequently verified by the authors, and we take full responsibility for the accuracy and appropriateness of all citations and related work discussions presented in this paper.

We also utilized LLMs to assist with paper writing, including improving grammar, enhancing clarity of explanations, and refining the presentation of our methodology and results. The LLMs were used as writing assistants to help articulate our ideas more clearly, but all technical content, experimental design, analysis, and conclusions remain the original intellectual contribution of the authors. We maintain full responsibility for all claims, representations, and technical content presented in this work, and have thoroughly verified all LLM-assisted content for accuracy and appropriateness.

