# OpenReview forum: "ResearchArcade: Graph Interface for Academic Tasks"
_ICLR.cc/2026/Conference — Submitted to ICLR 2026_

### Official Review · Reviewer_dTwN · 2025-10-31

**Soundness:** 2
**Presentation:** 2
**Contribution:** 3
**Rating:** 4
**Confidence:** 4

**Summary:**

This paper proposes ResearchArcade, a unified graph interface that integrates heterogeneous academic data from ArXiv and OpenReview. It uses a graph neural network to aggregate multi-source information (papers, reviews, figures, etc.) into node embeddings, which are then combined with textual inputs and processed by a large language model (Qwen3) for various academic tasks. Experiments show that incorporating graph-structured embeddings leads to consistent improvements over text-only baselines.

**Strengths:**

1. Heterogeneous information integration. The paper explores an interesting and valuable idea, using heterogeneous academic information (papers, reviews, figures, tables, revisions) to assist language-model–based prediction and generation tasks, which is a meaningful direction.

2. Clear pipeline design. The integration of graph neural networks and large language models is implemented in a clear, modular way: the GNN aggregates multi-source context into embeddings that are passed to the LLM, demonstrating a solid and technically sound pipeline design.

3.  Comprehensive empirical evaluation. The framework is comprehensive and empirically validated, covering six academic tasks and showing consistent, though moderate, improvements over text-only baselines, suggesting contribution to the community as a shared research infrastructure.

**Weaknesses:**

1. Limited methodological innovation.
The proposed Graph World Model (GWM) simply concatenates GNN-derived embeddings with textual inputs to the LLM, without introducing new learning mechanisms or joint optimization strategies.

2. Incomplete reproducibility.
The graph construction process, particularly the alignment between ArXiv and OpenReview papers is under-specified. There are no quantitative statistics on entity-matching accuracy, no discussion of handling multiple paper versions, duplicate author names, or conflicting metadata, and no validation against human-curated samples.

3. Shallow empirical analysis and lack of ablation.
While six tasks are included, the results are reported mainly as aggregate performance scores without in-depth ablation or interpretive analysis. The authors claim that incorporating graph structure improves performance, yet there is no investigation into which graph components or modalities (figures, tables, reviews) contribute to these gains.

4. Overstated “unified interface” claim.
Although positioned as a universal academic graph, each task is trained independently, and no cross-task transfer or shared representation is demonstrated. The claimed “unification” remains conceptual.

**Questions:**

Q1: How are the graph embeddings and text features integrated inside the LLM input? Are they concatenated at the embedding level or combined through an attention-based adapter?

Q2: How reliable is the mapping between ArXiv and OpenReview entities? Are there quantitative metrics validating the paper-to-paper alignment?

Q3: Does the framework support incremental updates as new papers and reviews are added, or is the graph static?

---

> ### Author Response · Authors · 2025-11-26
>
> Dear Reviewer dTwN
>
> We sincerely thank you for your careful and thorough review of our paper. Your insightful suggestions have been invaluable in improving the quality of our work.
>
> >"Summary: It uses a graph neural network to aggregate multi-source information (papers, reviews, figures, etc.) into node embeddings, which are then combined with textual inputs and processed by a large language model (Qwen3) for various academic tasks."
> "W1. Limited methodological innovation. The proposed Graph World Model (GWM) simply concatenates GNN-derived embeddings with textual inputs to the LLM, without introducing new learning mechanisms or joint optimization strategies."
>
> We thank the reviewer for their time and effort in reviewing our paper. However, we believe there may have been some misunderstandings in the summary of our contributions. We do not propose the _Graph World Model (GWM)_ [1]; as stated in _Section 5.1_, it is one of the base models used in our experiments.
>
> Instead, we proposed a unified data interface that supports various foundation models across a wide range of academic tasks as illustrated in _Figure 1_ and _Figure 2_. Previous work like _Pararev_[2], _DOCGENOME_[3], and _UARXIVE_[4] support only specific academic tasks and lack integration of graph structures. Although _OAG-Bench_ [5]  constructs distinct heterogeneous graphs for each academic tasks. Their scattered efforts limit scalability and supportability for novel academic tasks. In contrast, as shown in _Figure 1_, ResearchArcade integrates multi-source, multi-modal and temporal information into a unified heterogeneous graph. This framework is highly scalable since new nodes or edges can be simply added by appending new rows into tables. As mentioned in _Appendix A.2.2_, we support continuously crawling new academic data because the CRUD operations are efficient thanks to the PostgreSQL-backed design. Moreover, as stated in _Table 1_ and _Table 2_, our unified graph interface supports a wide range of academic tasks: 6 novel tasks, including citation prediction, paragraph generation, revision retrieval, revision generation, acceptance prediction, and rebuttal generation are implemented and 4 future tasks, such as idea generation, experiment planning, abstract writing, and review generation are proposed. Therefore, we believe ResearchArcade is a novel and valuable foundation for developing deep learning methods on diverse academic tasks.
>
> [1] Tao Feng, Yexin Wu, Guanyu Lin, and Jiaxuan You. Graph world model. 2025.
>
> [2] Leane Jourdan, Florian Boudin, Richard Dufour, Nicolas Hernandez, and Akiko Aizawa. Pararev: Building a dataset for scientific paragraph revision annotated with revision instruction. 2025.
>
> [3] Renqiu Xia, Song Mao, Xiangchao Yan, Hongbin Zhou, Bo Zhang, Haoyang Peng, Jiahao Pi, Daocheng Fu, Wenjie Wu, Hancheng Ye, et al. Docgenome: An open large-scale scientific document benchmark for training and testing multi-modal large language models. 2024.
>
> [4] Tarek Saier, Johan Krause, and Michael Farber. unarxive 2022: All arxiv publications pre-processed for nlp, including structured full-text and citation network. 2023.
>
> [5] Fanjin Zhang, Shijie Shi, Yifan Zhu, Bo Chen, Yukuo Cen, Jifan Yu, Yelin Chen, Lulu Wang, Qingfei Zhao, Yuqing Cheng, et al. OAG-Bench: a human-curated benchmark for academic graph mining. 2024.

---

> ### Author Response · Authors · 2025-11-26
>
> >"W2. Incomplete reproducibility. The graph construction process, particularly the alignment between ArXiv and OpenReview papers is under-specified. There are no quantitative statistics on entity-matching accuracy, no discussion of handling multiple paper versions, duplicate author names, or conflicting metadata, and no validation against human-curated samples."
>
> We thank the reviewer for pointing out these issue. To address the concerns, as stated in _Section 3.1_ (Connect ArXiv and OpenReview), we have confirmed that all the matched ArXiv paper and OpenReview submission have the same title, ensuring a 100% entity-mathcing accuracy. However, not all conferences submissions are uploaded to ArXiv, and about 45.43% of OpenReview submissions match to an ArXiv paper. Furthermore, we detailed the construction of ResearchArcade and discussed the data quality in _Appendix A.2.1_ and _Appendix A.2.3_. **Handling multiple paper versions**: We handle multiple paper versions easily in OpenReview, as they have distinct OpenReview IDs. For ArXiv papers, we specifically include the paper version in the paper table as a metadata. This is included in _Figure 1_, _Figure 3_ and and _Appendix A.2.1_. **Handling duplicate author names**: In OpenReview, authors with the same name are assigned different author IDs. In ArXiv, author identities are based on Semantic Scholar, where authors with the same name also have unique Semantic Scholar IDs. **Conflicting Metadata**: All metadata is collected from reliable academic platforms (ArXiv, OpenReview, Semantic Scholar), and we believe the metadata is non-conflicting. **Validation against Human-Curated Samples**: We have manually verified the crawled data, which matches the data accessed directly from these platforms.
>
> >"Q2: How reliable is the mapping between ArXiv and OpenReview entities? Are there quantitative metrics validating the paper-to-paper alignment?"
>
> We validate the paper-to-paper alignment based on the titles of ArXiv papers and OpenReview submissions. If the title of a paper/submission has changed, we assume that the content has undergone substantial changes and differs from the previous version. As stated in _Section 3.1_ (Connect ArXiv with OpenReview), we have confirmed that all the connected entities share the same title, ensuring the reliability of the mapping.
>
> >"W3. Shallow empirical analysis and lack of ablation. While six tasks are included, the results are reported mainly as aggregate performance scores without in-depth ablation or interpretive analysis. The authors claim that incorporating graph structure improves performance, yet there is no investigation into which graph components or modalities (figures, tables, reviews) contribute to these gains."
>
> We thank the reviewer for the suggestion and aggree that deeper abation studies would offer more valuable insights. As shown in _Table 4_, we conduct multimodal ablation studies on the Rebuttal Generation and Citation Prediction tasks across four settings: text-only, text-figures-only, text-tables-only, text-figures-tables. The results show that each modality (figures or tables) contributes to performance gains, and the best results are achieved by integrating both modalities. This confirms our conclusion that multimodal information is critical.
>
> Furthermore, we have added new ablation studies on review information for the Revision Retrieval and Revision Generation tasks across different models. For Revision Retrieval, review information consistently improves performance, especially for the EMB-based models. However, for revision generation, review information does not guarantee performance gains in smaller models, as they struggle to handle long-context queries. For larger models, review information brings only slight performance gains, likely due to the lack of explicit steps for revisions in most reviews.
>
> >"W4. Overstated “unified interface” claim. Although positioned as a universal academic graph, each task is trained independently, and no cross-task transfer or shared representation is demonstrated. The claimed “unification” remains conceptual."
>
> We thank the reviewer point out this claim, and we acknowledge that each task is trained independently. However, as shown in _Figure 1_, our proposed ResearchArcade serves as a unified data interface, enabling the implementation of six novel academic tasks and the proposal of four promising future tasks listed in _Table 3_ and _Table 4_. It is reasonable that we do not train a unified model to handle all tasks in this work due to the inherent difficulty of such task, and it deserves further independent exploration. And we do believe that our proposed ResearchArcade provides a solid foundation for the development of a unified model capable of addressing a wide range of academic tasks.

---

> ### Author Response · Authors · 2025-12-03
>
> >"Q1: How are the graph embeddings and text features integrated inside the LLM input? Are they concatenated at the embedding level or combined through an attention-based adapter?"
>
> For the Revision Retrieval task, as described in _Appendix A.6.3_, graph embeddings are concatenated at the embedding level inside the LLM input, following the _GWM_ [1] mechanism. For Paragraph Generation, Revision Generation, and Rebuttal Generation, as stated in Appendices A.6.2, A.6.4, and A.6.6, the graphs are integrated into the LLM input through textual descriptions.
>
> >"Q3: Does the framework support incremental updates as new papers and reviews are added, or is the graph static?"
>
> ResearchArcade supports incremental updates, allowing the graph to dynamically evolve as new data is incorporated. As illustrated in _Figure 1_, our heterogeneous graph is intuitively constructed based on the multiple node and edge tables in ResearchArcade. New nodes or edges can be simply added by appending new rows to these tables. Furthermore, as described in _Appendix A.2.2_, our PostgreSQL-backed design supports efficient CRUD operations and further enables the continous daily or weekly crawling of ArXiv data, resulting in a growing graph.
>
> [1] Tao Feng, Yexin Wu, Guanyu Lin, and Jiaxuan You. Graph world model. 2025.

---

### Official Review · Reviewer_iAbw · 2025-11-01

**Soundness:** 3
**Presentation:** 3
**Contribution:** 2
**Rating:** 4
**Confidence:** 3

**Summary:**

The paper presents ResearchArcade, a graph-based interface that connects ArXiv and OpenReview into a heterogeneous, temporally aware graph spanning text, figures, tables, reviews, and revisions. Tasks are defined in two steps: (1) pick a target entity and label, (2) retrieve the entity’s multi-hop neighborhood as input.

Based on this graph, authors introduce six tasks: figure/table insertion, paragraph generation, revision retrieval, revision generation, acceptance prediction, and rebuttal generation.

**Strengths:**

- Compared to previous datasets, ResearchArcade covers multiple sources and modalities. It also provides a unified interface for tasks defined on academic graphs.
- The writing is clear and easy to follow.

**Weaknesses:**

- As an academic graph, the dataset’s coverage of papers is still limited. It includes around 45k papers from arXiv and about 28k from ICLR, which may restrict the generalizability of conclusions derived from it.
- For the paragraph generation and revision generation tasks, the authors rely solely on semantic similarity metrics. However, such metrics may not capture aspects like clarity or appropriateness of the generated text (e.g., generated paragraphs and revisions). Using LLM-as-a-judge evaluations could better reflect quality.
- The paper would benefit from a more detailed discussion on why the proposed tasks are important and why they are worth studying within an academic graph framework. For instance, prior work exists for some of these tasks, such as revision generation (e.g., https://aclanthology.org/2025.wraicogs-1.4/). I think connecting to previous literature would help readers understand what new advantages graph-based approaches offer.

**Questions:**

1. How are the embeddings for figure nodes generated?

---

> ### Author Response · Authors · 2025-11-26
>
> Dear Reviewer iAbw
>
> We sincerely thank you for your careful and thorough review of our paper. Your insightful suggestions have been invaluable in improving the quality of our work.
>
> >"W1. As an academic graph, the dataset’s coverage of papers is still limited. It includes around 45k papers from arXiv and about 28k from ICLR, which may restrict the generalizability of conclusions derived from it."
>
> We thank the reviewer for this suggestion and agree that expanding the coverage of our dataset is important. To this end, as stated in _Section 3.1_, we have resulted in a dataset with 67k corpora from ArXiv in different scientific fields, including mathematics, physics, statistics, biology, economics, electrical engineering, astronomics, finance, as well as 57k submissions from EMNLP, ICML, NeurIPS and ICLR. The AAAI, ACL, ICCV, CVPR, ECCV, and IJCAI conferences are also investigated, but their data is not available. The detailed statistics are summarized in _Tables 5–13_ of _Appendix A.1_.
>
> Based on our updated dataset, we conduct the experiments on rebuttal generation and paragraph generation again and report the new results in _Table 3_ and _Table 4_. The original conclusions remain unchanged: the relational graph structures of ResearchArcade yields consistent gains and the multimodal information remains critical.
>
> >"W2. For the paragraph generation and revision generation tasks, the authors rely solely on semantic similarity metrics. However, such metrics may not capture aspects like clarity or appropriateness of the generated text (e.g., generated paragraphs and revisions). Using LLM-as-a-judge evaluations could better reflect quality."
>
> We thank the reviewer for pointing out this issue and agree that the semantic similarity metrics are insufficient and using LLM-as-a-judge evaluations is beneficial. As stated in _Section 5.1_ (Evaluation Metrics), we use _GPT-4o-mini_ [1] to perform the evaluation. The specific prompts for each generative tasks are listed in _Table 19-21_ of _Appendix A.6.1_, the evluation results are reported to _Table 3-4_, and the corresponding analysis are updated in _Section 5.2_ and _Appendix A.7_.
>
> >"W3. The paper would benefit from a more detailed discussion on why the proposed tasks are important and why they are worth studying within an academic graph framework. For instance, prior work exists for some of these tasks, such as revision generation (e.g., https://aclanthology.org/2025.wraicogs-1.4/). I think connecting to previous literature would help readers understand what new advantages graph-based approaches offer."
>
> We thank the reviewer for this suggestion and aggree that a more detailed discussion would strengthen our paper. To this end, for each academic task definition in _Section 4.2_, we have added a discussion on the task importance and the advantages of the academic graph framework compared to some related works.
>
> >"Q1. How are the embeddings for figure nodes generated?"
>
> As stated in _Section 5.1_ (Encoders), figure node embeddings are generated in two stages. First, a multimodal large language model generates a textual description. Then, the generated description is encoded into into embeddings using base model-specific encoders. While we experimented with _CLIP_[2], our current approach is more effective and easier to implement. This encoding framework is also flexible and can accommodate alternative multimodal encoders.
>
> [1] Aaron Hurst, Adam Lerer, Adam P Goucher, Adam Perelman, Aditya Ramesh, Aidan Clark, AJ Ostrow, Akila Welihinda, Alan Hayes, Alec Radford, et al. Gpt-4o system card. 2024.
>
> [2] https://openai.com/index/clip/

---

### Official Review · Reviewer_zZ9E · 2025-11-02

**Soundness:** 3
**Presentation:** 3
**Contribution:** 3
**Rating:** 6
**Confidence:** 4

**Summary:**

This paper introduces ResearchArcade, a graph-based interface designed to unify multi-source, multi-modal, and temporally evolving academic data for supporting a wide range of machine learning tasks in academic research. It integrates data from ArXiv and OpenReview, including text, figures, and tables, and models academic activities as heterogeneous graphs.

Key contributions include:
Unified graph interface for academic data across sources and modalities;
Two-step task definition scheme (identify target entity + retrieve neighborhood) to standardize academic tasks;
Six benchmark tasks (e.g., paragraph generation, revision retrieval, acceptance prediction) and four promising future tasks (e.g., idea generation, review generation);
Empirical validation showing that graph-based models outperform non-graph baselines, and multi-modal inputs improve performance;
Open-source release of data, pipeline, and evaluation protocols.

**Strengths:**

S1: This paper addresses a real need for unified academic data interfaces to support ML models across diverse research tasks.

S2: This paper integrates ArXiv and OpenReview with text, figures, tables, and temporal evolution, offering a holistic view of academic knowledge.

S3: The two-step scheme (target entity + neighborhood) of this paper is simple yet general, enabling both predictive and generative tasks.

S4: This paper evaluates 6 tasks across 4 model types (EMB, GNN, LLM, GWM), showing consistent gains from graph structure and multi-modal inputs.

S5: This paper releases data, code, prompts, and splits, making the work usable and extensible for future research.

S6: This paper includes a thoughtful ethics statement and reproducibility checklist, which is rare and commendable.

**Weaknesses:**

W1: The graph construction and task definition are engineering-heavy; no new modeling techniques or architectures are proposed.

W2: Tasks of this paper, like figure insertion or paragraph generation are reconstruction-style and may not reflect real-world academic needs (e.g., idea quality, scientific discovery).

W3: Best accuracy is only 0.55, barely above random — raises questions about whether the graph is rich enough for high-level reasoning.

W4: All metrics are automatic (SBERT, BLEU, etc.); no expert judgment on scientific validity, coherence, or usefulness of generated content.

W5: Data is CS-heavy (ArXiv + ICLR); no evaluation on how well the interface generalizes to other fields (e.g., biology, physics).

W6: This paper does not compare with larger SOTA LLMs (e.g., GPT-4, Claude, Qwen3-70B) or retrieve-augmented systems — limits external validity.

**Questions:**

Q1: How would ResearchArcade perform on open-ended scientific tasks like hypothesis generation, experiment design, or novel discovery?

Q2: Can the graph structure support more complex reasoning, such as causal chains, contradiction detection, or theory synthesis?

Q3: What is the human-perceived quality of generated paragraphs, rebuttals, or reviews? Are they scientifically sound or just fluent?

Q4: How scalable is the pipeline to larger corpora (e.g., PubMed, Semantic Scholar) or real-time updates (e.g., daily ArXiv dumps)?

Q5: Why does acceptance prediction fail even with rich graph data? Is it inherently unpredictable, or is the signal too weak?

Q6: How does ResearchArcade compare to retrieval-augmented LLMs or tool-augmented systems (e.g., ChatPDF, Elicit) on real user tasks?

---

> ### Author Response · Authors · 2025-11-26
>
> Dear Reviewer zZ9E
>
> We sincerely thank you for your careful and thorough review of our paper. Your insightful suggestions have been invaluable in improving the quality of our work.
>
> >"W1. The graph construction and task definition are engineering-heavy; no new modeling techniques or architectures are proposed."
>
> ResearchArcade is a unified data interface that supports various foundation models, rather than being a method tailored to specific tasks. Previous work like _Pararev_[1], _DOCGENOME_[2], and _UARXIVE_[3] support only specific academic tasks and lack integration of graph structures. Although _OAG-Bench_ [4] constructs distinct heterogeneous graphs for each academic tasks. Their scattered efforts limit scalability and supportability for novel academic tasks. In contrast, as shown in _Figure 1_, ResearchArcade integrates multi-source, multi-modal and temporal information into a unified heterogeneous graph. This framework is highly scalable since new nodes or edges can be simply added by appending new rows into tables. And as mentioned in _Appendix A.2.2_, we support continuously crawling new academic data because the CRUD operations are efficient thanks to the PostgreSQL-backed design. Moreover, as stated in _Table 1_ and _Table 2_, our unified graph interface supports a wide range of academic tasks: 6 novel tasks, including citation prediction, paragraph generation, revision retrieval, revision generation, acceptance prediction, and rebuttal generation, are implemented and 4 future tasks, such as idea generation, experiment planning, abstract writing, and review generation are proposed. Therefore, we believe ResearchArcade is a novel and valuable foundation for developing deep learning methods on diverse academic tasks.
>
> >"W2. Tasks of this paper, like figure insertion or paragraph generation are reconstruction-style and may not reflect real-world academic needs (e.g., idea quality, scientific discovery)."
>
> We thank the reviewer for pointing out this issue. For the Figure/Table Insertion task, we agree that it may reflect a limited real-world need (e.g., evaluating models' multimodal understanding). To address this concern, as described in _Section 4.2.1_, we have replaced it with Citation Prediction, a task that better aligns with real-world academic needs such as reference recommendation. For the Paragraph Generation task, as stated in _Section 4.2.2_, we believe it addresses the real-world need of assisting scientific writing.
>
> >"W3. Best accuracy is only 0.55, barely above random — raises questions about whether the graph is rich enough for high-level reasoning."
> "Q5: Why does acceptance prediction fail even with rich graph data? Is it inherently unpredictable, or is the signal too weak?"
>
> We thank the reviewer for highlighting these results. We believe that the Acceptance Prediction task is inherently difficult at this stage, since decisions depend on complex, subjective human judgments about novel scientific work ahead of time. As stated in the _Abstract_, our goal is not to claim that this task is currently solvable, but to “build a unified data interface to support the development of machine learning models for various academic tasks” whether or not fully solvable at the moment. Therefore, we view this task as exposing a meaningful open problem and our dataset as providing infrastructure for future work.
>
> Furthermore, even though the task is hard, graph-based models still outperform non-graph baselines, which suggests that the graph structure provides meaningful signal. We therefore view this task as highlighting an important open problem and our dataset as providing infrastructure for future work on this question.
>
>
> [1] Leane Jourdan, Florian Boudin, Richard Dufour, Nicolas Hernandez, and Akiko Aizawa. Pararev: Building a dataset for scientific paragraph revision annotated with revision instruction. 2025.
>
> [2] Renqiu Xia, Song Mao, Xiangchao Yan, Hongbin Zhou, Bo Zhang, Haoyang Peng, Jiahao Pi, Daocheng Fu, Wenjie Wu, Hancheng Ye, et al. Docgenome: An open large-scale scientific document benchmark for training and testing multi-modal large language models. 2024.
>
> [3] Tarek Saier, Johan Krause, and Michael Farber. unarxive 2022: All arxiv publications pre-processed for nlp, including structured full-text and citation network. 2023.
>
> [4] Fanjin Zhang, Shijie Shi, Yifan Zhu, Bo Chen, Yukuo Cen, Jifan Yu, Yelin Chen, Lulu Wang, Qingfei Zhao, Yuqing Cheng, et al. OAG-bench: a human-curated benchmark for academic graph mining. 2024.

---

> ### Author Response · Authors · 2025-11-26
>
> >"W4: All metrics are automatic (SBERT, BLEU, etc.); no expert judgment on scientific validity, coherence, or usefulness of generated content."
> "Q3: What is the human-perceived quality of generated paragraphs, rebuttals, or reviews? Are they scientifically sound or just fluent?"
>
> We thank the reviewer for this suggestion and agree that automatic metrics are insufficient for evaluating output quality. To this end, we include human analysis for Paragraph Generation, Review Generation, and Rebuttal Generation in _Appendix A.7.2, A.7.4_, and _A.7.6_, respectively. We also provide case studies on all the generation tasks in _Appendix A.8_. Overall, most generated outputs are fluent and coherent within the context. However, compared to the ground truth, the generated content may suffer from hallucinations, limiting their practical use in the real world.
>
> >"W5: Data is CS-heavy (ArXiv + ICLR); no evaluation on how well the interface generalizes to other fields (e.g., biology, physics)."
>
> We thank the reviewer for this suggestion and agree that expanding the coverage of our dataset is important. To this end, as stated in _Section 3.1_, we have added data from other scientific fields, including mathematics, physics, statistics, biology, economics, electrical engineering, astrophysics, and finance. In addition to ICLR, we also collect all the available peer-review data from EMNLP, ICML, and NeurIPS in OpenReview. The AAAI, ACL, ICCV, CVPR, ECCV, and IJCAI conferences are also investigated, but their data is not available. The resulting statistics are summarized in _Tables 5–13_ of _Appendix A.1_. However, it is reasonable that we initially focus on computer science field and ICLR conferences because they provide more open-source data. Resources for other scientific fields are relatively limited, especially regarding the peer-review data. And the reviews and revisions from most other conferences are not publicly available.
>
> Based on our updated dataset, we conduct the experiments on rebuttal generation and paragraph generation again and report the new results in _Table 3_ and _Table 4_. The original conclusions remain unchanged: the relational graph structures of ResearchArcade yields consistent gains and the multimodal information remains critical.
>
> >"W6: This paper does not compare with larger SOTA LLMs (e.g., GPT-4, Claude, Qwen3-70B) or retrieve-augmented systems — limits external validity."
> "Q6: How does ResearchArcade compare to retrieval-augmented LLMs or tool-augmented systems (e.g., ChatPDF, Elicit) on real user tasks?
>
> We thank the reviewer for pointing out these issues. To address the concerns, as described in _Section 5.1_ (Large Language models), we have added _GPTOSS-120B_ [1] as our baselines for the Revision Generation and Rebuttal Generation tasks and reported the results in _Table 3_ and _Table 4_. According to the results, the larger SOTA LLM outperforms smaller LLMs as expected.
>
> Furthermore, as shown in _Figure 4_ and _Figure 6_ of _Appendix A.8_, we compare ResearchArcade with _ChatPDF_ [2] on the Revision Retrieval and Rebuttal Generation tasks that reflect real user needs. The results indicate that while ChatPDF demonstrates a powerful PDF comprehending ability, the graph structures in ResearchArcade also provide significant value for paper understanding. Additionally, ResearchArcade proves to be a suitable framework for deep learning training, while the ChatPDF framework is not.
>
> [1] Sandhini Agarwal, Lama Ahmad, Jason Ai, Sam Altman, Andy Applebaum, Edwin Arbus, Rahul K Arora, Yu Bai, Bowen Baker, Haiming Bao, et al. gpt-oss-120b & gpt-oss-20b model card. 2025.
>
> [2] https://www.chatpdf.com/

---

> ### Author Response · Authors · 2025-11-26
>
> >"Q1: How would ResearchArcade perform on open-ended scientific tasks like hypothesis generation, experiment design, or novel discovery?"
> "Q2: Can the graph structure support more complex reasoning, such as causal chains, contradiction detection, or theory synthesis?"
>
> ResearchArcade can perform open-ended scientific tasks by leveraging reference answers from its universal academic graph, as shown in _Figure 3_ of _Appendix A.1_. For Hypothesis Generation, the hypothesis is defined by the paper’s conclusion (paragraphs connected to the conclusion section), with generation signals derived from the paper's references. For Experiment Design, the experiment is defined by the paper's experiment section, with signals extracted from the abstract (to infer the research question and method). For Novel Discovery, the novelty of an idea (e.g., the paper's method) is defined by its similarity score to existing ideas in the collected papers.
>
> ResearchArcade can also handle the tasks require complex reasoning becuase some edges in its academic graph represent causal relationship between nodes. For Causal Chains, the revision-review edges capture causal links between reviews and revisions. For Contradiction Detection, revisions that replace ideas or facts signal potential contradictions, which can trace back the specific review that identified the issue. For Theory Synthesis, the paper-paper edge represents citation relationships, and using generation signals from cited papers, the task is to synthesize ideas in the citing paper.
>
> Furthermore, to facilitates the definitions of this tasks, as mentioned in _Section 3.1_ (LLM-Generated Content), we leverage LLMs for further data preprocessing and append new columns to the openreview_revisions and arxiv_papers tables. Specifically, we classify the revisions according to the categories described in [1] and extract the research questions and methods for each paper primarily from its abstract.
>
> >"Q4: How scalable is the pipeline to larger corpora (e.g., PubMed, Semantic Scholar) or real-time updates (e.g., daily ArXiv dumps)?"
>
> ResearchArcade offers strong scalability and supports real-time updates. As illustrated in _Figure 1_, our heterogeneous graph is intuitively constructed based on the multiple node and edge tables in ResearchArcade. New nodes or edges from larger corpora (e.g., PubMed, Semantic Scholar) can be simply added by appending new rows to these tables. Furthermore, as described in _Appendix A.2.2_, our PostgreSQL-backed design supports efficient CRUD operations and further enables the continous daily or weekly crawling of ArXiv data.
>
> [1] L´eane Jourdan, Florian Boudin, Richard Dufour, Nicolas Hernandez, and Akiko Aizawa. Pararev: Building a dataset for scientific paragraph revision annotated with revision instruction. 2025.

---

### Official Review · Reviewer_NgQe · 2025-11-03

**Soundness:** 2
**Presentation:** 3
**Contribution:** 2
**Rating:** 4
**Confidence:** 3

**Summary:**

This submission proposes RESEARCHARCADE, a graph-based interface that integrates multi-source data—computer science papers from ArXiv and peer reviews/submissions from OpenReview—and multi-modal information (text, figures, and tables) to support the development of machine learning models for diverse academic tasks. It adopts a two-step scheme—“identify target entity and retrieve neighborhood”— to unify the definitions of six academic tasks(e.g., Figure/Table Insertion, Paragraph Generation) and is compatible with various models, including embedding-based models (EMB, e.g., Longformer), graph neural networks (GNN, e.g., HANConv). However, critical resources are unavailable in this submission: no accessible code, related dataset samples, or key preprocessing details are provided. The authors only commit to releasing these materials upon publication. Furthermore, this work functions more like a data preprocessing pipeline than an innovative contribution—no novel technical approaches are proposed.

**Strengths:**

S1. This submission accurately identifies key pain points in academic AI. From a data perspective, it targets the complexity of academic data, sourced from diverse platforms (ArXiv’s computer science papers, OpenReview’s ICLR submissions) and spanning multiple modalities (text, figures, tables). From a task perspective, it decreases the unnecessary effort required for data. By focusing on these gaps, the work directly responds to the demand for a unified data interface, as proposed in its core design.

S2. The heterogeneous graph structure effectively encompasses diverse entities and their relationships. Entities include not only papers, reviews, figures, and tables but also ArXiv paragraphs and OpenReview revisions. Key relationships are explicitly modeled: e.g., "paper-paragraph" and "paper-figure/table".

S3. The framework demonstrates good adaptability to different model types. It supports embedding-based models, graph neural networks, large language models, and the Graph World Model. This compatibility enables it to handle both predictive tasks and generative tasks.

**Weaknesses:**

W1. Data used in the experiment is restricted to the computer science (CS) field (ArXiv data) and ICLR conferences (OpenReview data). However, no testing was conducted in other domains such as biology, chemistry or materials science.

W2. Critical preprocessing steps are not explained. For example, how to extract paragraphs and figures from ArXiv’s LaTeX and how to align OpenReview reviews with paper paragraphs—these details remain untackled.

W3. Novel academic-related contributions are limited, such as mechanisms for rapidly updating the graph when new paper revisions are added.

**Questions:**

1. Do you plan to conduct cross-domain validation in some non-CS fields?

2. Could you add academic-oriented optimizations (e.g., methods to resolve ambiguity in OpenReview review comments)?

---

> ### Author Response · Authors · 2025-11-26
>
> Dear Reviewer NgQe
>
> We sincerely thank you for your careful and thorough review of our paper. Your insightful suggestions have been invaluable in improving the quality of our work.
>
> >"Summary: However, critical resources are unavailable in this submission: no accessible code, related dataset samples, or key preprocessing details are provided."
>
> To address the concerns, the critical resources are currently available.
>
> Specifically, our full codebase is accessible at https://anonymous.4open.science/r/research-arcade-E815/, including the code to reproduce our framework and dataset samples for all the components of our heterogeneous academic graph. We did not release these resources at the initial submission stage as most ICLR submissions also choose not to share their code publicly at that time.
>
> Furthermore, we have detailed the specific preprocessing process in _Appendix A.2_. We show step-by-step details to construct our academic graph from ArXiv and OpenReview in _Appendix A.2.1_ and _Appendix A.2.3_, separately.
>
> >"Summary: Furthermore, this work functions more like a data preprocessing pipeline than an innovative contribution—no novel technical approaches are proposed."
>
> ResearchArcade is a unified data interface that supports various foundation models, rather than being a method tailored to specific tasks. Previous work like _Pararev_[1], _DOCGENOME_[2], and _UARXIVE_[3] support only specific academic tasks and lack integration of graph structures. Although _OAG-Bench_[4] constructs distinct heterogeneous graphs for each academic tasks, their scattered efforts limit scalability and supportability for novel academic tasks. In contrast, as shown in _Figure 1_, ResearchArcade integrates multi-source, multi-modal and temporal information into a unified heterogeneous graph. This framework is highly scalable since new nodes or edges can be simply added by appending new rows into tables. And as mentioned in _Appendix A.2.2_, we support continuous crawling new academic data because the CRUD operations are efficient thanks to the PostgreSQL-backed design. Moreover, as stated in _Table 1_ and _Table 2_, our unified graph interface supports a wide range of academic tasks: 6 novel tasks, including citation prediction, paragraph generation, revision retrieval, revision generation, acceptance prediction, and rebuttal generation, are implemented and 4 future tasks, such as idea generation, experiment planning, abstract writing, and review generation, are proposed. Therefore, we believe that ResearchArcade is a novel and valuable foundation for developing deep learning methods on diverse academic tasks.
>
> [1] Leane Jourdan, Florian Boudin, Richard Dufour, Nicolas Hernandez, and Akiko Aizawa. Pararev: Building a dataset for scientific paragraph revision annotated with revision instruction. 2025.
>
> [2] Renqiu Xia, Song Mao, Xiangchao Yan, Hongbin Zhou, Bo Zhang, Haoyang Peng, Jiahao Pi, Daocheng Fu, Wenjie Wu, Hancheng Ye, et al. Docgenome: An open large-scale scientific document benchmark for training and testing multi-modal large language models. 2024.
>
> [3] Tarek Saier, Johan Krause, and Michael Farber. unarxive 2022: All arxiv publications pre-processed for nlp, including structured full-text and citation network. 2023.
>
> [4] Fanjin Zhang, Shijie Shi, Yifan Zhu, Bo Chen, Yukuo Cen, Jifan Yu, Yelin Chen, Lulu Wang, Qingfei Zhao, Yuqing Cheng, et al. OAG-Bench: a human-curated benchmark for academic graph mining. 2024.

---

> ### Author Response · Authors · 2025-11-26
>
> >"W1. Data used in the experiment is restricted to the computer science (CS) field (ArXiv data) and ICLR conferences (OpenReview data). However, no testing was conducted in other domains such as biology, chemistry or materials science."
> "Q1. Do you plan to conduct cross-domain validation in some non-CS fields?"
>
> We thank the reviewer for this suggestion and agree that expanding the coverage of our dataset is important. To this end, we have added data from other scientific fields, including mathematics, physics, statistics, biology, economics, electrical engineering, astronomics, and finance. In addition to ICLR, we also collect all the available peer-review data from EMNLP, ICML, and NeurIPS in OpenReview. The AAAI, ACL, ICCV, CVPR, ECCV, and IJCAI conferences are also investigated, but their data is not available. The resulting statistics are summarized in Tables 5–13 of _Appendix A.1_. However, it is reasonable that we initially focus on computer science field and ICLR conferences because they provide more open-source data. Resources for other scientific fields are relatively limited, especially regarding the peer-review data. And the reviews and revisions from most other conferences are not publicly available.
>
> Based on our updated dataset, we conduct experiments on rebuttal generation and paragraph generation again and report the new results in _Table 3_ and _Table 4_. The original conclusions remain unchanged: the relational graph structures of ResearchArcade yields consistent gains and the multimodal information remains critical.
>
> >"W2. Critical preprocessing steps are not explained. For example, how to extract paragraphs and figures from ArXiv’s LaTeX and how to align OpenReview reviews with paper paragraphs—these details remain unresolved."
>
> We thank the reviewer for pointing out this issues. To address the concerns, the detailed preprocessing pipeline for LaTeX source code from ArXiv is described in _Appendix A.2.1_. The main-content processing stages include Paper Source File Downloading, Author Information Processing, Section-level Decomposition, Paragraph-level Decomposition, Citation Information Processing, Figure/Table Extraction, and Graph Construction and Storage. Furthermore, _Appendix A.2.3_ provides details on preprocessing peer-review data from OpenReview, where reviews are aligned with paper paragraphs via review–paper and paper–paragraph edges.
>
> >"W3. Novel academic-related contributions are limited, such as mechanisms for rapidly updating the graph when new paper revisions are added."
>
> We believe that the ResearchArcade’s scalability mechanism itself constitutes a novel academic-related contribution. As illustrated in _Figure 1_, our heterogeneous graph is intuitively constructed based on the multiple node and edge tables in ResearchArcade. New nodes or edges (e.g., paper revisions) can be simply added by appending new rows to these tables, thereby updating the graph. Furthermore, as described in _Appendix A.2.2_, our PostgreSQL-backed design  supports efficient CRUD operations and further enables the continuous daily or weekly crawling of ArXiv data.
>
> >"Q2. Could you add academic-oriented optimizations (e.g., methods to resolve ambiguity in OpenReview review comments)?"
>
> We thank the reviewer for their thoughtful feedback. In the revised version, we added several academic-oriented optimizations in _Appendix A.2.4_, including Revision Classification, Research Question Summarization, and Method Summarization. These generated columns support additional academic tasks and help models process the data more efficiently.
>
> Regarding the example of resolving ambiguity in OpenReview comments, while we acknowledge that this task is a crucial task (many submissions suffer from vague OpenReview comments), we argue that this task itself cannot be rigorous defined and evaluated: there is no clear definition of what constitutes ambiguity in review comments, nor labeled data indicating when a review is “clear” versus “unclear” or when a clarification is “good.” Without such a task definition and labels, it is difficult to design and fairly evaluate this component. With a more precise definition and appropriate annotation, we are confident that our framework and dataset could be extended to include such an analysis in future work.

---

### Author Response · Authors · 2025-11-26

| **Dimension**                          | **Reviewer NgQe**                          | **Reviewer zZ9E**                                | **Reviewer iAbw**                      | **Reviewer dTwN**                                        | **Action / Response**                                                                                                                                                                                          |
| -------------------------------------- | ------------------------------------------ | ------------------------------------------------ | -------------------------------------- | -------------------------------------------------------- | ---------------------------------------------------------------------------------------------------------------------------------------------------------------------------------------------------------------------------- |
| **Evaluations**                 | —                                          | “What is the human-perceived quality of generated paragraphs, rebuttals, or reviews?”                 | “Using LLM-as-a-judge evaluations could better reflect quality.”                   | —                                                        | **Action**: Added **GPT-4o-mini LLM-as-judge** evaluation (_Table 3-4_); included human case studies for generation tasks (_Appendix A.7–A.8_).                                                                                              |
| **Presentation**         | —                                          | —                                                | “The paper would benefit from a more detailed discussion on why the proposed tasks are important and why they are worth studying within an academic graph framework.” | —                                                        | **Action**: Added **expanded related work** and **justification of graph advantages** for revision and generation tasks. (_Section 4.2.1-Section 4.2.6_)                                                                                                        |
| **Others**         | “Need ambiguity resolution in reviews.”   | “Accuracy in Acceptance Prediction barely above random.” Difficulty   | - | —  | **Action**: For Reviewer NgQe, we added **Revision Classification, Research Question Summarization, Method Summarization** (_Appendix A.2.4_); For Reviewer zZ9E, we acknowledged the inherent difficulty of this task, given the subjective nature of scientific evolution. |

Summary: We substantially expanded dataset scope, added cross-domain experiments, strengthened reproducibility, introduced human and LLM-based evaluation, added ablations and SOTA baselines, and clarified the novelty and scalability of ResearchArcade as a unified academic graph infrastructure.

---

### Author Response · Authors · 2025-11-26

| **Dimension**                          | **Reviewer NgQe**                          | **Reviewer zZ9E**                                | **Reviewer iAbw**                      | **Reviewer dTwN**                                        | **Action / Response**                                                                                                                                                                                          |
| -------------------------------------- | ------------------------------------------ | ------------------------------------------------ | -------------------------------------- | -------------------------------------------------------- | ---------------------------------------------------------------------------------------------------------------------------------------------------------------------------------------------------------------------------- |
| **Scalability**              | —                                          | “How scalable is the pipeline.”                           | —                                      | “Is the graph static?”                                       | **Action**: ResearchArcade is backed by **PostgreSQL**, supports daily/weekly crawling and simple **row-level graph updates** for incremental expansion. (_Appendix A.2.2_)                                                                           |
| **Task Design**                   | —                                          | “Tasks are are reconstruction-style and may not reflect real-world academic needs.”  “Can it supports open-ended and complex reasoning tasks?”        | —                                      | —                                                        | **Action & Response**: Replaced Figure/Table Insertion with **Citation Prediction** (_Section 4.2.1_); Defined formal input/output interfaces for new tasks proposed by reviewers.                                                                              |
| **Experiments** | “No cross-domain validation in non-CS fields.”                                          | “Does not compare with larger SOTA LLMs and tool-augmented systems” | —                                      | “No investigation into which graph components or modalities contribute to these gains.”                                                        | **Action**: Added experiments on **cross-domain datasets** (_Table 3_); added **GPT-OSS-120B** baselines (_Table 3-4_); conducted **ChatPDF case studies** (_Appendix A.8_); conducted **new ablation studies** on multimodal and review information. (_Table 4_)                                                                           |

---

### Author Response · Authors · 2025-11-26

We sincerely thank the reviewers for their time and thoughtful feedback. Below, we summarize the major points raised across reviews. The “Action / Response” column provides the actions taken and the rebuttal responses for each point.

| **Dimension**                          | **Reviewer NgQe**                          | **Reviewer zZ9E**                                | **Reviewer iAbw**                      | **Reviewer dTwN**                                        | **Action / Response**                                                                                                                                                                                          |
| -------------------------------------- | ------------------------------------------ | ------------------------------------------------ | -------------------------------------- | -------------------------------------------------------- | ---------------------------------------------------------------------------------------------------------------------------------------------------------------------------------------------------------------------------- |
| **Reproducibility**      | “No accessible code or dataset samples.” “How to extract ... from ArXiv’s LaTeX and how to align OpenReview reviews with paper paragraphs ... remain untackled.”   | —                                                | —                                      | “Incomplete reproducibility. How reliable is the mapping between ArXiv and OpenReview entities?”                            | **Action**:  Full codebase and dataset samples are released via public link; preprocessing details are included in _Appendix A.2_ with step-by-step ArXiv/OpenReview pipelines.                                                            |
| **Novelty & Contribution**     | “More like a data preprocessing pipeline than an innovative.” | “Engineering-heavy, no new modeling.”            | —                                      | “Limited methodological innovation.” “Unified claim overstated; tasks trained independently.”                    | **Response**: ResearchArcade is a **unified, scalable data interface** (not a task-specific model); emphasized novelty in **multi-source, multi-modal, temporal heterogeneous graph** and table-based **extensibility**. |
| **Dataset Coverage**                   | “Data restricted to the computer science (CS) field (ArXiv data) and ICLR conferences.”                     | “Data is CS-heavy (ArXiv + ICLR).”                    | “Dataset’s coverage of papers is still limited.”              | —                                                        | **Action**: Crawled and integrated **math, physics, biology, statistics, economics, EE, astronomy, finance**; added EMNLP, ICML, NeurIPS; added statistics in _Tables 5–13_. Conducted experiments on new dataset. (_Table 3_)                                                  |

---

### Author Response · Authors · 2025-12-03

# Continued General Response

| Dimension | Key Concerns | Our Main Actions | Summary |
| --- | --- | --- | --- |
| **Coverage & Scalability** | 1）Initially appears limited to CS/ICLR; 2）unclear whether the graph can grow or stay up-to-date. | 1）Expanded to ~67k ArXiv papers across multiple fields and ~57k OpenReview submissions from ICLR, ICML, NeurIPS, EMNLP (_Section 3.1_); 2）described the support for a continuous crawling pipeline for new papers and submissions, enabled by the multi-table and PostgreSQL-backed design (_Appendix A.2.2_); | ResearchArcade is now a **multi-field, multi-venue resource** with a clear mechanism for **sustained expansion**, making it a durable community benchmark rather than a static dataset. |
| **Tasks, Experiments & Evaluations** | 1) Some tasks are not aligned with real user needs; 2) requested cross-domain validation, stronger baselines, and deeper ablations; 3) lack of human analysis and LLM-based evaluation. | 1) Replaced figure/table insertion with **citation prediction**, and clarified how all tasks map to real user needs in writing and review workflows (_Section 4.2.1-4.2.6_); 2) Added **cross-domain experiments** (different scientific domains & venues), a **stronger LLM baseline** (GPT-OSS-120B), a **tool-augmented baseline** (ChatPDF), **multimodal/review ablations** (_Table 3-4, Appendix A.8_); 3) Added GPT-4o-mini as **LLM-as-judge**, and **human analysis** on examples (_Appendix A.7_). | The final task designs **match real user needs in academic workflows**; the results show that **graph structure and multimodal context consistently improves performance across domains and tasks**; and the evaluation is substantially stronger and more informative for future method development. |
___
### Rebuttal Conclusion

During the rebuttal process, We addressed all areas of concern in revision, as summarized in the concise table above. The revised paper now demonstrates:
- A clearly defined heterogeneous, extensible, unified academic graph interface with **public code/data and mature pipelines**.
- Expanded **multi-field, multi-venue coverage**, supporting academic tasks across diverse scenarios.
- Improved **task design and representation**, clarify the real user needs underlying each task, and explain why a graph structure benefits each of them.
- Deeper experiments were conducted, with **stronger baselines, cross-domain validation, and more thorough ablations**. The results show consistent benefits from graph and multimodal structure and enabling rigorous future work on research assistants and scientific reasoning.
- Detailed evaluation, including **human analysis and LLM-as-a-judge assessments**, is provided to capture the clarity, appropriateness, and human-perceived quality of generated paragraphs, rebuttals, and reviews.

However, due to the information leakage incident, the reviewers were unable to respond or update their evaluations. We therefore provide a detailed analysis of the initial scores and argue that the revised paper merits acceptance.

* Reviewer zZ9E gave our work **an initial score of 6 with a confidence of 4**, indicating that our work is **above the acceptance threshold with high confidence**.
* Reviewers NgQe and iAbw each gave **an initial score of 4 with a confidence of 3**. Considering the actions and responses we have provided to address their comments, **it is highly likely they would have increased their scores to an acceptance level**.
* Reviewer dTwN gave **an initial score of 4 with a confidence of 4**. However, the summary of our paper from this reviewer indicates a fundamental misunderstanding of our work’s contributions. While some constructive feedback was offered, **the validity of the score is questionable**.

All in all, given the strengths of our work and the scope of our targeted revisions, we believe the paper’s contribution merits acceptance.

Best regards,

The Authors

---

### Author Response · Authors · 2025-12-03

# General Response

Dear Area Chair,

We understand the constraints of the current review process. We are encouraged that all reviewers recognized the paper's core strengths and appreciate the constructive feedback they provided. We have fully addressed all their concerns in our revision and believe that our paper merits acceptance.
___

### Summary of Contribution
This paper proposes **ResearchArcade**, a **heterogeneous**, **scalable**, and **unified** data interface with **graph structures**, to support machine learning models across diverse academic tasks. We extensively evaluated ResearchArcade on six novel academic tasks and found that incorporating graph structures **consistently improves performance**.

___
We summarize the highlights below to assist in your final assessment.
### Summary of Strengths

Despite the variance in scores, all reviewers show agreement on several core strengths of our work:

- **Substantive Infrastructure Contribution:** ResearchArcade utilizes a coherent multi-table format to organize academic data as a heterogeneous graph, providing strong scalability.（Reviewer NgQe, zZ9E)
- **Comprehensive Data Coverage:** ResearchArcade integrates **multi-source, multi-modal and temporal** information across diverse scientific domains and venues.（Reviewer NgQe, zZ9E, iAbw, dTwN)
- **Diverse and Realistic Task Coverage:** ResearchArcade provides a unified framework supporting multiple predictive and generative academic tasks. We implemented 6 novel tasks and proposed 4 future tasks, all align with real writing and review workflows. （Reviewer NgQe, zZ9E, iAbw, dTwN)
- **Strong Compatibility with Various Models:** ResearchArcade supports various models, including embedding-based, GNN-based, LLM-based, and GWM-based models.（Reviewer NgQe, zZ9E)
- **Meaningful Empirical Insights:** Extensive experiments (6 tasks across 4 base models) show the **consistent benefits of incorporating graph structure** into academic tasks.（Reviewer NgQe, zZ9E)
- **Good Paper Representation:** Our paper is well-written and easy-to-follow.（Reviewer iAbw)

### Resolution of Key Concerns with Actions

We concisely summarize the reviewers' main concerns and our corresponding actions below.

| Dimension | Key Concerns | Our Main Actions | Summary |
| --- | --- | --- | --- |
| **Contribution** | 1) Is this mostly a preprocessing pipeline? 2) What is new beyond prior academic graphs/benchmarks? 3) Confusion about GWM as our main contribution. | 1) Clarified that **ResearchArcade is a heterogeneous, extensible, unified academic graph interface**, not a new model. We **implemented 6 novel tasks and proposed 4 future tasks**; 2) We compare our work with related works (e.g., OAG-Bench, Pararev, UNARXIVE, and DOCGENOME) to highlight its novelty and contribution (_Section 2_); 3) GWM is a baseline model, not our contribution (_Response to dTwN_). | Our contribution is now clearly framed as a **data interface**: a heterogeneous, extensible, unified and graph-based data interface for various academic tasks, complementary to methodological work and distinct from existing datasets. |
| **Reproducibility** | 1) No public code/data samples; 2) missing details for ArXiv/OpenReview processing; 3) unclear reliability of ArXiv–OpenReview mapping. | 1) Released an anonymized **full codebase and dataset samples** (_Response to NgQe_); 2) added **step-by-step pipeline descriptions** for ArXiv, OpenReview, and continuous crawling (_Appendix A.2.1, A.2.3_); 3) documented the reliable title-based paper alignment and its coverage (approximately 45%) (_Section 3.1_). | The data interface now can be **reconstructed** from the codebase and the data samples are available. The Arxiv/OpenReview data processing pipelines, along with their connections, are **transparent and reliable**. |

---

### Meta-Review · Area_Chair_iEqm · 2026-01-06

**Summary:**

As a precursor to this summary of concerns, I should mention that deciding the final rating for this paper was extremely hard, given that I believe in the merits of collecting additional datasets that are "living," i.e., that can be dynamically updated as time progresses. That being said, I also believe that this work merits another round of revisions and potentially even a change of venue like [DMLR](https://data.mlr.press) or the NeurIPS Datasets & Benchmarks track. This will ensure that reviewers already have a background in dataset generation and curation since the discussion demonstrated that numerous questions concerning these aspects arose and could only be partially addressed by the authors.

Next to the concerns raised by reviewers (see below), I also have some personal concerns about this work, specifically the number of _design choices_ that went into it. The sheer amount of different tools involved even at the data collection level make this a daunting project (and the authors are to be commended for their efforts!), with the data collection alone already comprising seven steps. However, it is unclear to what extent these steps have been tested; the accompanying code contains no tests and some brief checks of the example CSVs show that the "Section" assignment does not appear to work reliably:

In the `arxiv_paragraphs.csv`, I find the following excerpt:

```
id,paragraph_id,content,paper_arxiv_id,paper_section,section_id,paragraph_in_paper_id
23484735,901,\begin{lemma} If each $\Lambda_i$ in $\vec{\Lambda} = \{\Lambda_{i}\}_{1 \leq i \leq k}$, $k \geq 2$, is $f$-checkable, then $\bigoplus_{\vec{d_{0}}}  \vec{\Lambda}$  is also $f$-checkable. \end{lemma},2504.20637,Lingos,293222,1040
```
This paragraph belongs to the following preprint: [Protocol Dialects as Formal Patterns: A Composable Theory of Lingos -- Technical report
](https://arxiv.org/abs/2504.20637), where it appears to be associated to [Section 4: Lingo Compositions](https://arxiv.org/pdf/2504.20637#page=20), since it appears at [page 21 of the PDF](https://arxiv.org/pdf/2504.20637#page=21). However, as we can see from the excerpt, it is assigned to the wrong section. Of course, such errors _can_ happen and I do not want to claim that this invalidates the whole data collection; nevertheless, in a project of this size, having code available is not enough; there is a need for rigorous testing and data quality assurance.

At the risk of sounding nitpicky, I want to mention that this is crucial because the very foundation of the constructed graphs relies on making the _right_ choices during data collection. My simpler proposal would be to first present a thoroughly-tested version of the collection and crawling process, including the separation into individual paragraphs first, followed by a careful assessment of _how_ to model graphs on these data. I am raising this point because it is a priori not clear that the ways the graphs are constructed right now is the "right" way or the "best" way. As such, I cannot endorse the paper for publication now.

My verdict was also reached by considering the following concerns by reviewers, which have only been partially addressed:

- Preprocessing steps not sufficiently explained (reviewer `NgQe`): While the code is now available, there are still numerous choices and dependencies, making it hard to assess the overall correctness of the pipeline even _before_ graph construction.

- Unclear alignment of tasks with academic needs (reviewer `zZ9E`): While partially addressed by introducing a _new_ task, viz., citation prediction, it is still unclear whether this is the "right" task one should look. As the authors mention themselves, it is not clear whether these tasks are at all relevant for all disciplines equally, and, furthermore, each task relies on yet another set of external databases like [SemanticScholar](https://www.semanticscholar.org/), making data quality even harder to assess.

- Unclear alignment of graph with tasks (reviewer `zZ9E`): It is not clearly established to what extent a graph is required _at all_ for accomplishing these tasks. Moreover, there are numerous ways of constructing a graph from such data, and this is also not investigated by the authors.

- Insufficient discussion of relevance of tasks (reviewer `iAbw`): While the authors added a brief discussion on the tasks here, I find this to be insufficient to explain _why_ it should be possible to solve some of them. Even the "acceptance prediction" is exacerbated by unclear targets, since one may asks "Accepted at _first_ try at original submission venue" or "Accepted in _some_ form at possible another venue." I get the impression that these tasks are hard to define in practice.

- Unclear reproducibility and missing ablations (reviewer `dTwN`): While this was partially mitigated by showing the code, the code raises even more questions as to the technical choices. Moreover, ablations remain only a small part of the revision, without specifying to what extent individual choices in preprocessing affect the results.

I thus believe that further revisions are warranted and I hope the authors can take this actionable feedback to improve their work.

**Reviewer Concerns:**

Many reviewers raised concerns about the *reproducibility* of the work. As I outlined above, I do not believe that such concerns have been alleviated by releasing untested (!) code. The concerns that were addressed in the rebuttal were largely clustered around adding more tasks and providing clarifying questions on the collection pipeline. The major fundamental concerns on the construction of the graphs from data _per se_ remain, however.

**Reviewer Scores:**

- Reviewer `NgQe`: Initial score: 4; expected score: 4. I do not believe that the concerns of the reviewer wrt. reproducibility and design choices have been adequately addressed.

- Reviewer `zZ9E`: Initial score: 6; expected score: 8. Some of the concerns of the reviewer have been addressed; hence, erring on the side of caution it is possible that the reviewer might raise their score.

- Reviewer `iAbw`: Initial score: 4; expected score: 4. Not all concerns, in particular those pertaining to the tasks themselves, have been addressed.

- Reviewer `dTwN`: Initial score: 4; expected score: 4. The concerns on reproducibility have not been addressed.

According to my understanding, even in the best-case scenario, only a single reviewer would have raised their score, without possibly being willing to champion the paper (given that they self-report a lower confidence). I thus believe that this submission would have remained a borderline submission, requiring a careful final verdict, which I hope to have provided.

---

### Decision · Program_Chairs · 2026-01-26

Reject